# Treatment of MRSA-infected osteomyelitis using bacterial capturing, magnetically targeted composites with microwave-assisted bacterial killing

Yuqian Qiao[1], Xiangmei Liu [2✉], Bo Li[3], Yong Han[3], Yufeng Zheng [4], Kelvin Wai Kwok Yeung[5], Changyi Li[6], Zhenduo Cui[1], Yanqin Liang[1], Zhaoyang Li[1], Shengli Zhu[1], Xianbao Wang [2] & Shuilin Wu [1✉]

Owing to the poor penetration depth of light, phototherapy, including photothermal and photodynamic therapies, remains severely ineffective in treating deep tissue infections such as methicillin-resistant *Staphylococcus aureus* (MRSA)-infected osteomyelitis. Here, we report a microwave-excited antibacterial nanocapturer system for treating deep tissue infections that consists of microwave-responsive $Fe_3O_4$/CNT and the chemotherapy agent gentamicin (Gent). This system, $Fe_3O_4$/CNT/Gent, is proven to efficiently target and eradicate MRSA-infected rabbit tibia osteomyelitis. Its robust antibacterial effectiveness is attributed to the precise bacteria-capturing ability and magnetic targeting of the nanocapturer, as well as the subsequent synergistic effects of precise microwaveocaloric therapy from $Fe_3O_4$/CNT and chemotherapy from the effective release of antibiotics in infection sites. The advanced target-nanocapturer of microwave-excited microwaveocaloric-chemotherapy with effective targeting developed in this study makes a major step forward in microwave therapy for deep tissue infections.

[1] School of Materials Science & Engineering, The Key Laboratory of Advanced Ceramics and Machining Technology by the Ministry of Education of China, Tianjin University, Tianjin 300072, China. [2] Hubei Key Laboratory of Polymer Materials, Ministry-of-Education Key Laboratory for the Green Preparation and Application of Functional Materials, School of Materials Science & Engineering, Hubei University, Wuhan 430062, China. [3] State Key Laboratory for Mechanical Behavior of Materials, School of Materials Science and Engineering, Xi'an Jiaotong University, Xi'an 710049 Shanxi, China. [4] College of Engineering, State Key Laboratory for Turbulence and Complex System, Department of Materials Science and Engineering, Peking University, Beijing 100871, China. [5] Department of Orthopaedics & Traumatology, Li Ka Shing Faculty of Medicine, The University of Hong Kong, Pokfulam, Hong Kong 999077, China. [6] Stomatological Hospital, Tianjin Medical University, Tianjin 300070, China. ✉email: liuxiangmei1978@163.com; shuilinwu@tju.edu.cn

Osteomyelitis, a deep tissue infection, is well known to cause abscesses, organ infection, and sepsis, and is life threatening[1,2]. It is estimated that the incidence of bone infections after open fractures exceeds 30% and the median cost for osteomyelitis inpatients between 2013 and 2016 was US $10,504[3], which may be higher now. Clinically, osteomyelitis is often treated by systemically injecting a large amount of antibiotics, which entails long treatment and recovery times[4]. The overuse of antibiotics has been reported to cause drug-resistance in clinical settings[5,6]. Large doses of antibiotics can also impair the innate immune system and cause serious adverse reactions, including fever, kidney damage, and thrombophlebitis[7]. Methicillin-resistant *Staphylococcus aureus* (MRSA) is often found to be the causative pathogenic organism in cases of osteomyelitis[8,9]. In order to tackle the crisis of antibiotic resistance, both antibiotic-free and antibiotic-strengthening strategies are being developed, such as phototherapy and photo-assisted antibiotics therapy. However, owing to the poor penetration depth of near infrared light, such strategies are only viable for subcutaneous tumors[10,11] or wound infections[12]; they are not suitable for treating deep tissue infection. Therefore, it is urgent to develop better therapies that can effectively treat deep osteomyelitis, including MRSA infections, with minimum antibiotic toxicity.

Compared to solar light, microwave (MV), as an electromagnetic spectrum, has deep penetration ability, high microwaveocaloric therapy (MCT) efficiency, and negligible side effects, which make it a promising candidate in clinical settings[13–15]. In particular, the MCT was enhanced by adjusting the impedance matching and attenuation constant between magnetic and dielectric materials of microwaveocaloric sensitivity agents, which can absorb MV energy and convert it into thermal energy. Che et al. developed the enhanced coupling of the electronic field to facilitate the chemical activity of the magnetic nanoparticles[16,17] and pioneer the investigation of chemical reaction mechanisms in the nanometer devices[18]. Magnetic iron oxide ($Fe_3O_4$) nanoparticles not only have large magnetic loss[19], and thus produce huge magnetico calor under MV excitation, but also have already been approved by the US Food and Drug Administration for medical use[20]. Carbon nanotubes (CNTs), as a dielectric material, haves good electrical contact and fast electron conduction, exhibiting great potentials to be an excellent MV absorbent[21,22].

However, neither dielectric nor magnetic microwaveocaloric sensitizers have excellent absorption ability when they are utilized alone[23]. Nanohybrids of magnetic and dielectric sensitizers, with tailored dielectric and magnetic loss as well as obvious interfacial polarization induced by sufficient interfacial interactions[24], are promising high-performance MCT agents.

To minimize the adverse effects of hyperthermia induced by MV, an effective strategy of eradicating deep tissue infection is to combine MCT with chemotherapy. An active bacteria-targeting nanocapturer with controlled release of antibiotics in particular not only reduces the toxicity of antibiotics but also improves therapeutic effects for bacterial infections[9]. This bacteria-targeting capability can also be achieved by promoting interaction between nanoagents and bacteria via their chemical components on the surface (e.g., proteins, peptidoglycan, and lipopolysaccharides)[25,26]. It has been reported that physical forces such as magnetic fields can deliver drug-loaded nanoplatform to infection sites with high specificity and efficiency[27]. Meanwhile, phase change material (PCM), e.g., 1-tetradecanol (which has a melting point of ~40 °C), as a new type of functional material, can quickly respond to temperature fluctuation and can be transformed into liquid, which is helpful for controlling the release of therapeutic drugs by tailoring the temperature of the nanomachine[28].

Here we report a dual-targeting and MV-excited drug release system for the synergistic eradication of MRSA-induced osteomyelitis using a combined microwaveocaloric-chemotherapy (MCCT) system (Fig. 1). In this system, mesoporous $Fe_3O_4$ nanospheres are combined with functionalized CNTs to produce a neural-network-like structure of $Fe_3O_4$/CNT, which demonstrated better impedance matching and optimized attenuation constant, thus making it an efficient sensitizer for MCT. Subsequently, gentamicin (Gent) loaded into the synthesized $Fe_3O_4$/CNT nanocomposites for chemotherapy is introduced, together with 1-tetradecanol (as a PCM) to control the release of Gent. The final nanocomposites ($Fe_3O_4$/CNT/Gent) rapidly and efficiently eradicated MRSA-induced osteomyelitis by capturing the bacteria and enabling the MCCT to subsequently kill it in combination with magnetic targeting. This strategy is promising for improving the penetration of microwaveocaloric sensitizers and enhancing bacteria-specific synergistic therapies.

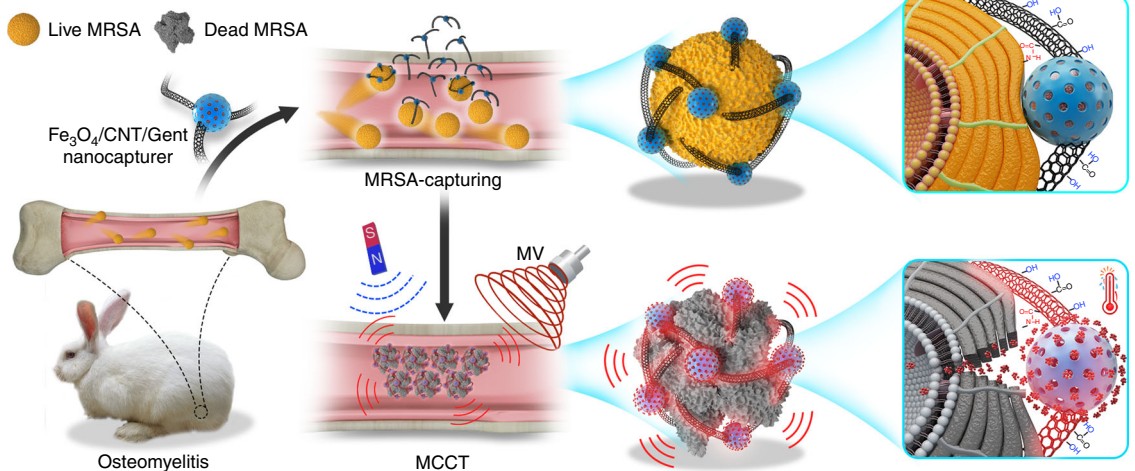

**Fig. 1 The schematic illustration of the microwaveocaloric-chemotherapy (MCCT) of $Fe_3O_4$/CNT/Gent.** The CNT displayed on the nanocapturer potently captured MRSA. When microwave-excited, the nanocapturers of $Fe_3O_4$/CNT are activated to generate thermal energy, which can locally melt PCM (the melting point is ~40 °C) and then locally release drug (Gent) at the site of the osteomyelitis with MRSA infection, finally eradicating MRSA precisely by combined the MCCT and an external magnetic field.

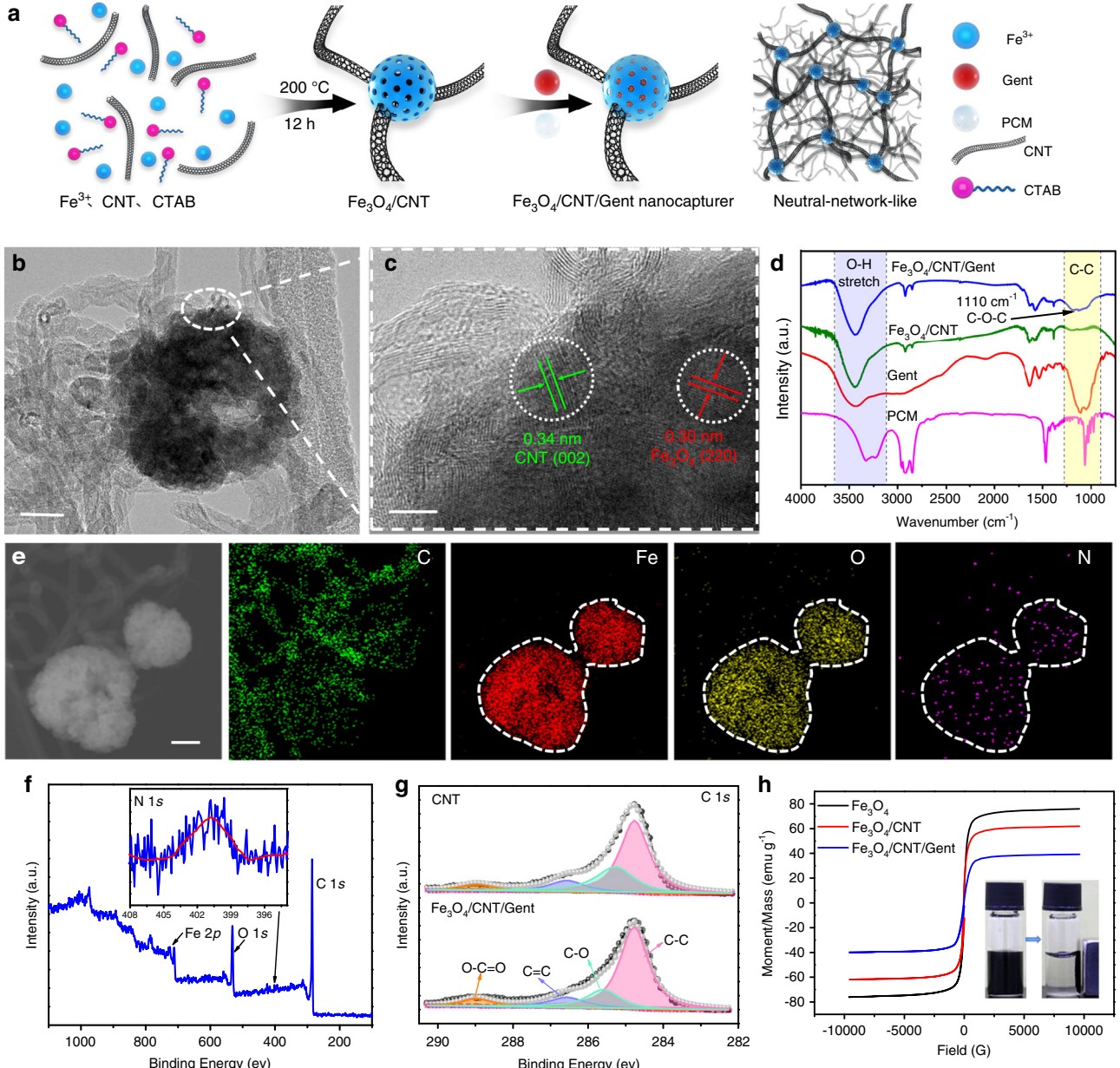

**Fig. 2 Characterization of the synthesized Fe₃O₄/CNT/Gent nanocapturer. a** Schematic illustration of the preparation procedures of the Fe₃O₄/CNT/Gent nanocapturer. Typical TEM (**b**) and HRTEM (**c**) images of the Fe₃O₄/CNT/Gent nanocapturer. Scale bar, 25 nm (**b**) and 5 nm (**c**). **d** FTIR spectra of the Gent, PCM, Fe₃O₄/CNT, and Fe₃O₄/CNT/Gent. **e** TEM element mapping of Fe₃O₄/CNT/Gent. Scale bar, 50 nm. **f** The XPS curves of Fe₃O₄/CNT/Gent. Inset: magnified XPS curves of N 1s. **g** The C 1s XPS spectra in CNT and Fe₃O₄/CNT/Gent. **h** The magnetic properties of Fe₃O₄, Fe₃O₄/CNT, and Fe₃O₄/CNT/Gent. Inset: the photograph showing the ferrofluidic behavior of Fe₃O₄/CNT/Gent in physiological saline. Neodymium rare earth permanent magnet with grade N38 magnetic energy, around 30 mm × 20 mm × 10 mm in length, width, and height, respectively. Source data are provided as a Source Data file.

## Results

**Synthesis and characterization.** The Fe₃O₄/CNT samples were prepared using the one-pot solvothermal method (Fig. 2a). In this process, CNTs (Supplementary Fig. 1) were dispersed homogeneously and partly threaded with Fe₃O₄ nanospheres as neurites to form a neural-network-like structure (Fig. 2a), in which Fe₃O₄ nanospheres acted as somas in the structure and uniformly adhered to the overlapping sections of the assembled CNTs (Supplementary Fig. 2). This unique structure of Fe₃O₄/CNT is benefited to improve the thermal conversion efficiency under MV owing to its multiple reflections. The X-ray diffraction (XRD)

pattern of the Fe₃O₄/CNT (Supplementary Fig. 3) exhibits almost the same peaks as pure Fe₃O₄ except for the peak at $2\theta = 26°$, which was assigned to the (002) plane of the graphite-like structure of CNTs[29], indicating the successful combination of CNTs and Fe₃O₄ nanospheres. The N₂ adsorption–desorption curves (Supplementary Fig. 4) suggested that the Fe₃O₄/CNT had a high specific surface area of 133.5 m² g⁻¹ due to the presence of abundant mesopores, with the average diameters centered at 3 and 45 nm (Supplementary Fig. 4, inset curve). The Gent was then loaded into mesoporous Fe₃O₄/CNT and encapsulated by PCM through physical shaking to obtain the Fe₃O₄/CNT/Gent

nanocapturers (Fig. 2b and Supplementary Fig. 5). Obvious heterointerfaces between $Fe_3O_4$ nanospheres and CNTs were observed from the high-resolution transmission electron microscopy (HRTEM) of $Fe_3O_4$/CNT/Gent (Fig. 2c), which was beneficial for accumulating more free charges, thus producing Debye relaxation to convert MV into thermal energy[30]. The $Fe_3O_4$ nanospheres and CNTs were further determined by well-defined lattice planes with interplanar distances of 0.30 and 0.34 nm, corresponding to the (220) lattice planes of $Fe_3O_4$[31], and the (002) lattice planes of CNT[32], respectively. The Fourier-transform infrared spectroscopy (FTIR) of $Fe_3O_4$/CNT/Gent compared to $Fe_3O_4$/CNT, the obvious absorption peaks in the FTIR obtained from the $Fe_3O_4$/CNT/Gent nanocapturer confirmed that the Gent and PCM were successfully loaded onto the $Fe_3O_4$/CNT (Fig. 2d): the peak at 1110 $cm^{-1}$ was attributed to the vibration of C–O–C in the Gent; the absorption band between 3630 and 3100 $cm^{-1}$ corresponded to the O–H stretching vibration; and that between 1250 and 900 $cm^{-1}$ was related to the alicyclic chain vibrations of C–C, which originated from PCM. UV–vis absorbance spectra (Supplementary Fig. 6) further confirmed the successful loading of Gent in the $Fe_3O_4$/CNT. Both the element mappings (Fig. 2e) and the X-ray photoelectron spectroscopy (XPS) full spectrum (Fig. 2f) synergistically confirmed the compositions of $Fe_3O_4$/CNT/Gent with the C, Fe, O, and N elements, and the N element showed a similar distribution area as the Fe and O elements according to the mappings of $Fe_3O_4$/CNT/Gent, proving the uniform loading of Gent in mesoporous nanospheres. The C 1s spectrum was deconvoluted into four peaks, corresponding to graphitized carbon with various functional groups, as shown in Fig. 2g. A clear positive shift (~0.3 eV) of C–O binding energy was observed on the high-resolution spectrum of C 1s for $Fe_3O_4$/CNT/Gent compared to the CNT, further suggesting a strong chemical interaction between $Fe_3O_4$ nanospheres and CNTs. The zeta potential of $Fe_3O_4$/CNT (Supplementary Fig. 7) changed from −13.145 to +21.015 mV after loading Gent and PCM. The $Fe_3O_4$/CNT/Gent in fetal bovine serum did not precipitate after setting one hour (Supplementary Fig. 8), which proved that this hybrid was relatively stable under complex physiological conditions.

We measured the magnetic properties of the prepared $Fe_3O_4$/CNT/Gent. As shown in Fig. 2h, the saturation magnetization value of $Fe_3O_4$ was 76.0 emu $g^{-1}$, suggesting that its magnetic response property was strong. In contrast, the saturation magnetization value of $Fe_3O_4$/CNT (61.9 emu $g^{-1}$) and $Fe_3O_4$/CNT/Gent (39.2 emu $g^{-1}$) decreased, due to the diamagnetic nature of the CNT, Gent, and PCM. Although the measured saturation magnetization value of $Fe_3O_4$/CNT/Gent was lower than that of the pristine $Fe_3O_4$, it was enough to guarantee its rapid separation from aqueous solution within 3 s under an external magnet, which enabled its biological collection and separation. Significantly, the $Fe_3O_4$/CNT/Gent nanocapturer could still be completely attracted by an extra magnet even behind 16-mm-thick pork tissue (Supplementary Fig. 9), although it required a longer time (~4 min) in thicker pork tissue, indicating the deep penetration ability of the magnetic field to attract these prepared nanocapturers. And the biodegradation behavior of $Fe_3O_4$/CNT/Gent is discussed in the Supplementary Fig. 10.

**Mechanism of enhanced microwaveocalortic effect and the thermo-responsive performance.** According to the evolution of $Fe_3O_4$/CNT/Gent temperature with MV power densities (Supplementary Fig. 11), we chose a medical low-power-intensity (2.45 GHz, 0.1 W $cm^{-2}$) to avoid higher temperatures adversely affecting ambient healthy tissues. As shown in Fig. 3a, under MV

irradiation, the temperature of $Fe_3O_4$/CNT and $Fe_3O_4$/CNT/Gent rose as high as 55.5 and 52.8 °C, respectively, within 5 min, while under the same conditions, the control group (physiological saline), $Fe_3O_4$, and CNT only reached the low temperatures of 44.5, 45.5, and 46.3 °C, respectively. Furthermore, the microwaveocalortic conversion efficiency ($\eta$) of $Fe_3O_4$/CNT dispersed in saline solution is 35.7% (Supplementary Table 1). We found that $Fe_3O_4$/CNT/Gent could retain a stable microwaveocalortic effect after three cycles (Supplementary Fig. 12), indicating that the microwaveocaloric stability of the nanocapturers was favorable. The synthesized $Fe_3O_4$/CNT/Gent nanocapturers thus had a desirable microwaveocaloric effect and MV stability, making them a promising candidate for MCT.

We investigated the thermo-responsive Gent release characteristics of $Fe_3O_4$/CNT/Gent nanocapturers under MV excitation. The drug loading efficiency of $Fe_3O_4$/CNT/Gent is 8.89%. After applying MV to the $Fe_3O_4$/CNT/Gent solution for 20 min at a preset time, the Gent was rapidly released from the nanocapturer; a high Gent release ratio of more than 81.5% was achieved after five times of MV activations (Fig. 3b). In contrast, the nanocapturer released little Gent (31.6%) without MV treatment (Ctrl) after 48 h, which was observed from the platform of the leaching curve. These results confirmed that the microwaveocaloric effect generated by $Fe_3O_4$/CNT under MV excitation melted PCM to effectively trigger and control the rapid release of Gent from the $Fe_3O_4$/CNT/Gent.

To elucidate the microwaveocaloric mechanism of the nanocapturer as prepared, the reflection loss (RL) value and the electromagnetic parameters were characterized. Generally, the RL spectrum reflects the MV absorption ability of materials, so a lower RL value means more MV has been absorbed[24], suggesting excellent microwaveocaloric effect. As shown in Fig. 3c, at the frequency of 2.45 GHz, the RL values of the CNT and the $Fe_3O_4$ were −1.18 dB (~23.79% absorption) and −1.78 dB (~33.63% absorption), respectively, indicating their lower MV absorption ability, which was similar to previous reports[23]. A significant increase of MV absorption was achieved after the $Fe_3O_4$ was combined by CNTs and the RL value decreased to −9.84 dB (~89.62% absorption) at 2.45 GHz. After being loaded with Gent and PCM, the corresponding RL value remained at −9.51 dB (88.81% of MV absorption), suggesting that the MV absorption ability of the synthesized nanocapturer is much stronger than previously reported antitumor MV sensitizers (25.76% absorption)[33]. This strong MV absorption performance simultaneously requires both suitable impedance matching ($|Z_{in}/Z_0|$) and a desirable attenuation constant ($\alpha$)[24]. Good impedance matching is a prerequisite for the good MV absorption performance of materials. Furthermore, a high $\alpha$ value is another important factor for materials to achieve high MV absorption performance. Ideal impedance matching requires a $|Z_{in}/Z_0|$ value of 1.0, which implies the complete entrance of the incident electromagnetic wave into the absorbent[19], and a higher $\alpha$ value means that more entranced MV is absorbed by converting them into heat. Both $Fe_3O_4$/CNT and $Fe_3O_4$/CNT/Gent exhibited better $|Z_{in}/Z_0|$ values (0.5265 and 0.5158, respectively), indicating that they were subjected to more incident MV entranced into them than $Fe_3O_4$ (0.3540) or CNT (0.0702) (Fig. 3d). They also showed the balanced $\alpha$ values between $Fe_3O_4$ and CNT (Fig. 3e), indicating that the CNT enhanced the dissipating capability of MV energy—in other words, the CNT enhanced the MV absorption ability of hybrids by dissipating MV energy into heat energy. In contrast, the CNT had the lowest $|Z_{in}/Z_0|$ value (Fig. 3d) and $Fe_3O_4$ nanospheres possessed the lowest $\alpha$ value (Fig. 3e) among them, suggesting that the incident MV was hindered by the worst impedance matching of CNT and the lowest MV conversion efficiency of $Fe_3O_4$. Consequently, both CNT and $Fe_3O_4$ exhibited

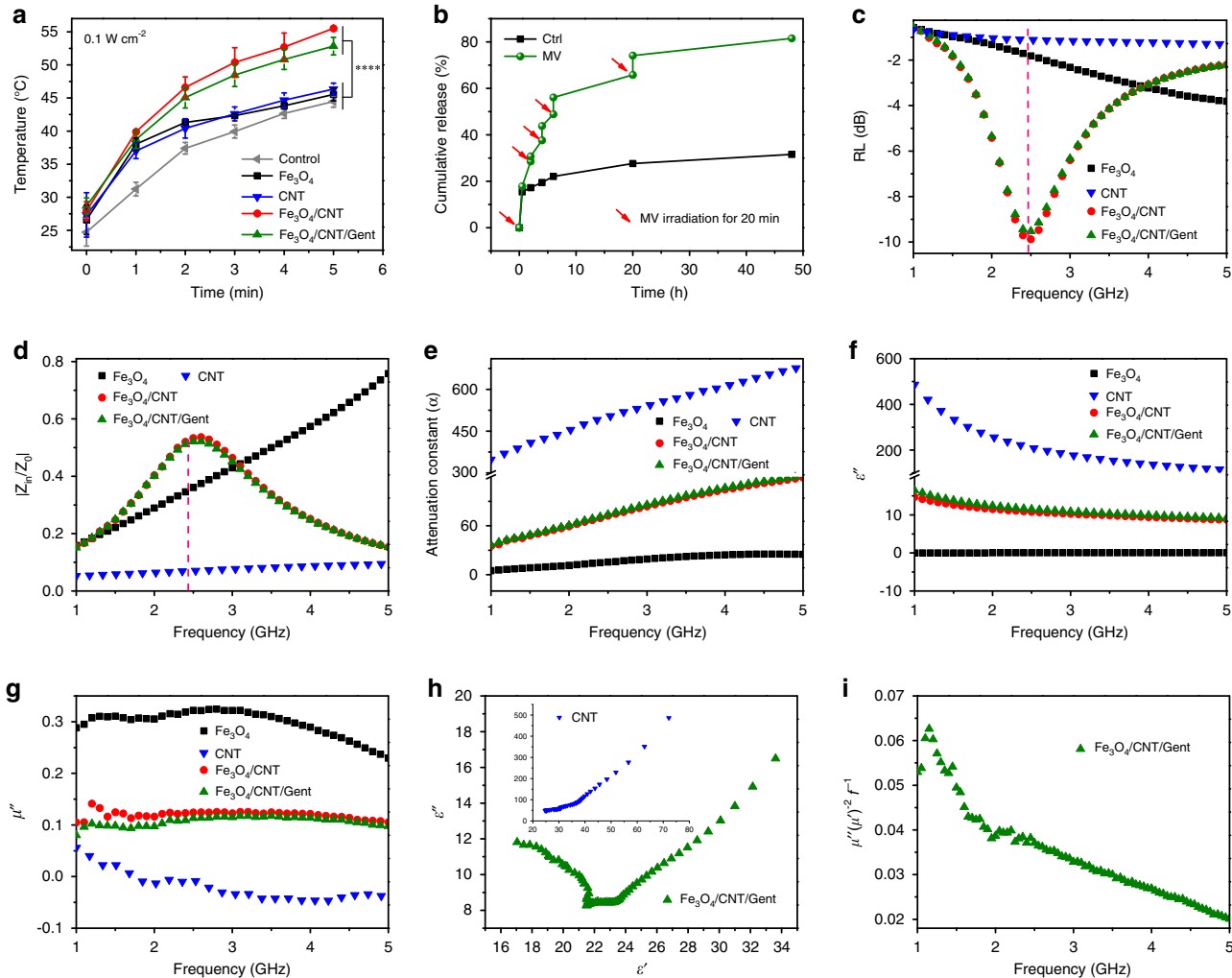

**Fig. 3 Mechanism of enhanced microwaveocaloric effect and the thermo-responsive performance. a** The microwaveocaloric curves of the $Fe_3O_4$, CNT, $Fe_3O_4$/CNT, and $Fe_3O_4$/CNT/Gent under MV excitation for 5 min. Data are presented as mean ± standard deviations from a representative experiment ($n = 3$ independent samples). ****$P < 0.0001$ analysed by two-way ANOVA with Tukey's multiple comparisons post hoc test. **b** In vitro Gent release profile from $Fe_3O_4$/CNT/Gent under MV excitation or not. **c** Reflection loss curves of $Fe_3O_4$, CNT, $Fe_3O_4$/CNT, and $Fe_3O_4$/CNT/Gent. **d** Frequency dependence of relative input impedance ($|Z_{in}/Z_0|$). **e** Frequency dependence of attenuation constant α. Imaginary part of complex permittivity (**f**) and permeability (**g**) of the four samples in the 1–5 GHz range. **h** Plots of $\varepsilon'$ versus $\varepsilon''$ for $Fe_3O_4$/CNT/Gent. Inset: plots of $\varepsilon'$ versus $\varepsilon''$ for CNT. **i** Frequency dependence of $\mu''$ $(\mu')^{-2}$ $f^{-1}$ values for $Fe_3O_4$/CNT/Gent. Source data are provided as a Source Data file.

high RL values and poor microwaveocaloric effects. Meanwhile, the unique neural-network-like structure of $Fe_3O_4$/CNT/Gent was beneficial because it effectively absorbed MV energy by promoting multiple scattering when MV was propagated in the network of hybrids (Supplementary Fig. 13).

Specifically, both $Fe_3O_4$/CNT and $Fe_3O_4$/CNT/Gent showed dielectric loss (Fig. 3f) and magnetic loss (Fig. 3g). The dielectric loss of a material is related to polarization relaxation and conduction loss. Polarization relaxation behaviors can be illustrated by the relationship between $\varepsilon'$ and $\varepsilon''$ curve according to the Eq. (1)[19]:

$$(\varepsilon' - \varepsilon_\infty)^2 + (\varepsilon'')^2 = (\varepsilon_s - \varepsilon_\infty)^2. \quad (1)$$

As shown in Fig. 3h, the $\varepsilon' - \varepsilon''$ curve of $Fe_3O_4$/CNT/Gent exhibits obvious large Cole–Cole semicircles compared to the CNT, demonstrating that several Debye relaxations processes occur in $Fe_3O_4$/CNT/Gent. Interfacial polarization was considered the dominant polarization mode in composite absorbers due to the different permittivity and conductivity between $Fe_3O_4$ and

CNT and more free charges were accumulated at the heterointerface of $Fe_3O_4$/CNT/Gent (Supplementary Fig. 13), leading to the enhanced interfacial polarization that produced Debye relaxation that absorbed MV[19]. Thanks to the high electron conductivity of CNT, conductive loss, which was proportional to the electron conductivity, was another important source for dielectric loss, and the surface functional groups of CNT provided the dipole polarization that increased dielectric loss[34]. Hence, MV quickly attenuated in the numerous conductive networks and was converted into heat. $Fe_3O_4$ nanoparticles induced magnetic loss in the hybrid to dissipate MV, as illustrated in Fig. 3g. The eddy current loss was analyzed using the $\mu''$ $(\mu')^{-2}$ $f^{-1}$ curve to evaluate its contribution to magnetic loss. If the eddy current effect occurs in $Fe_3O_4$/CNT/Gent, the calculated $\mu''$ $(\mu')^{-2}$ $f^{-1}$ will be a constant value[19]. As shown in Fig. 3i, the decrease of $\mu''$ $(\mu')^{-2}$ $f^{-1}$ values with frequency dependence demonstrated that the eddy current was suppressed, and the natural ferromagnetic resonance domination became the main source for magnetic loss within the experimental frequency region.

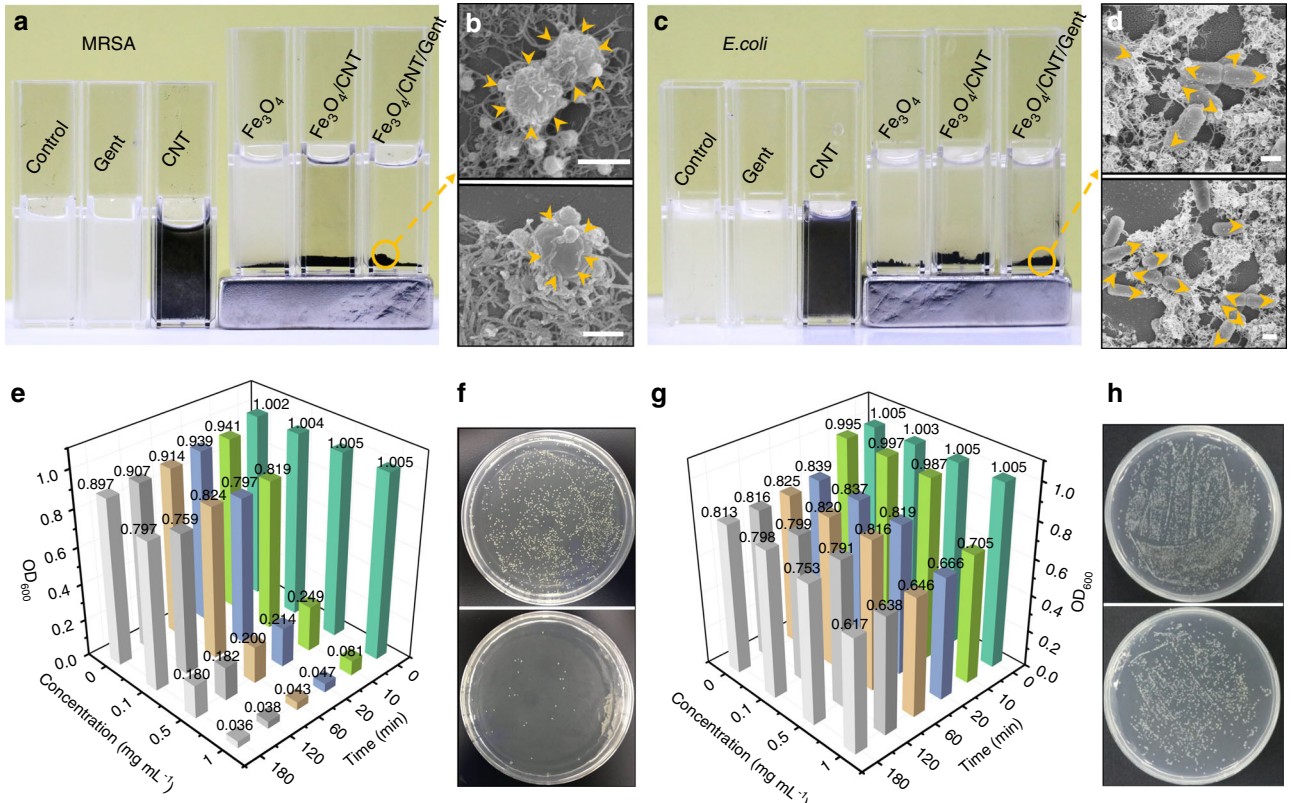

**Fig. 4 The bacteria-capturing ability of Fe₃O₄/CNT/Gent nanocapturer. a–d** Capturing MRSA (**a**) and *E. coli* (**c**) by different samples. Neodymium rare earth permanent magnet with grade N38 magnetic energy. Representative SEM observation of Fe₃O₄/CNT/Gent binding to MRSA (**b**) and *E. coli* (**d**). The orange arrows in **b**, **d** indicate the Fe₃O₄/CNT/Gent attached to bacteria. Scale bar, 500 nm. **e–h** The overall capturing bacteria viability as a function of time and Fe₃O₄/CNT/Gent concentration for MRSA (**e**) and *E. coli* (**g**). OD value is positive correlated with the amounts of bacteria in the solution. The spread plate of MRSA (**f**) and *E. coli* (**h**) before (above) and after (below) dispersing Fe₃O₄/CNT/Gent. $n = 1$ independent sample for **e**, **g**. Source data are provided as a Source Data file.

To summarize, Fe₃O₄/CNT has higher microwaveocalortic effect than Fe₃O₄ or CNT alone due to the good impedance matching and reasonable attenuation constant (mainly includes heterointerfaces polarization and promoted multiple scattering). Specifically, the good impedance matching and interfacial polarization promoted the multiple scattering, conductive loss, dipole polarization, and natural ferromagnetic resonance loss, thereby greatly enhancing the microwaveocaloric performance of the Fe₃O₄/CNT.

**Bacteria-capturing ability**. As shown in Fig. 4, when the Fe₃O₄/CNT/Gent or Fe₃O₄/CNT nanoparticles were dispersed into the suspensions of Gram-positive MRSA or Gram-negative *Escherichia coli* (*E. coli*), they could capture these two kinds of bacteria, which was evident from the clear supernatant (Fig. 4a, c). In contrast, paramagnetic Fe₃O₄ did not show such efficient bacteria-capturing ability, as was evident from the turbid supernatant, which was the similar as the pure bacterial dispersions. First, we ruled out the electrostatic-binding as the main origin for bacteria-capturing, because both the bacteria and the Fe₃O₄/CNT are all negatively charged. Furthermore, supposing that electrostatic force is responsible for binding between Fe₃O₄/CNT/Gent and bacteria, the positively charged Fe₃O₄ should exhibit strong binding with bacteria rather than the nonbinding ability shown in Fig. 4a, c. Based on these results, we proposed that the excellent bacteria-capturing ability of Fe₃O₄/CNT should be attributed to the existing of CNT. However, limited by its nonmagnetic properties for CNT, its capturing ability is not apparent in the images presented Fig. 4a, c.

To clarify this, we prepared Fe₃O₄/CNT hybrids with different CNT amounts (Supplementary Fig. 14) and tested the capturing activity of these hybrids to MRSA. As shown in Supplementary Fig. 15, as the amount of CNT added was increased, the MRSA-capturing capacity of hybrids became stronger with a lower optical density (OD) value. However, when the amount of CNT added reached 1.5 times that of Fe₃O₄ (mass ratio), the OD value for Fe₃O₄/CNT₂₂₅ increased compared to Fe₃O₄/CNT₁₅₀. Too many CNTs being added may have induced an increasing number of free-CNTs that were not completely combined with Fe₃O₄. These free-CNTs could not be attracted by the magnetic field, thus leading to an increase in the OD value. These results suggested that CNT played a key role in capturing bacteria.

Different capturing patterns of Fe₃O₄/CNT/Gent to MRSA and *E. coli* were subsequently observed by SEM. The Fe₃O₄/CNT/Gent homogeneously covered the surface of MRSA (Fig. 4b), but preferentially attached to the two poles of *E. coli* (Fig. 4d). The Fe₃O₄/CNT/Gent also exhibited a better ability to capture MRSA than *E. coli*. The overall OD value of both MRSA (Fig. 4e) and *E. coli* (Fig. 4g) diminished with time and with increasing Fe₃O₄/CNT/Gent concentration, and the MRSA group decreased faster than the *E. coli* group, suggesting that the Fe₃O₄/CNT/Gent-captured MRSA more efficiently. It is clear in the spread plate that the ability of Fe₃O₄/CNT/Gent (1 mg mL⁻¹, 20 min) to capture MRSA (95.32%) (Fig. 4f) is stronger than its ability to capture *E. coli* (33.73%) (Fig. 4h). The capturing behavior of Fe₃O₄/CNT/Gent being different between these two kinds of bacteria was due to the different composition on the surface of the Gram-positive (MRSA) and Gram-negative (*E. coli*) bacteria[35]. In

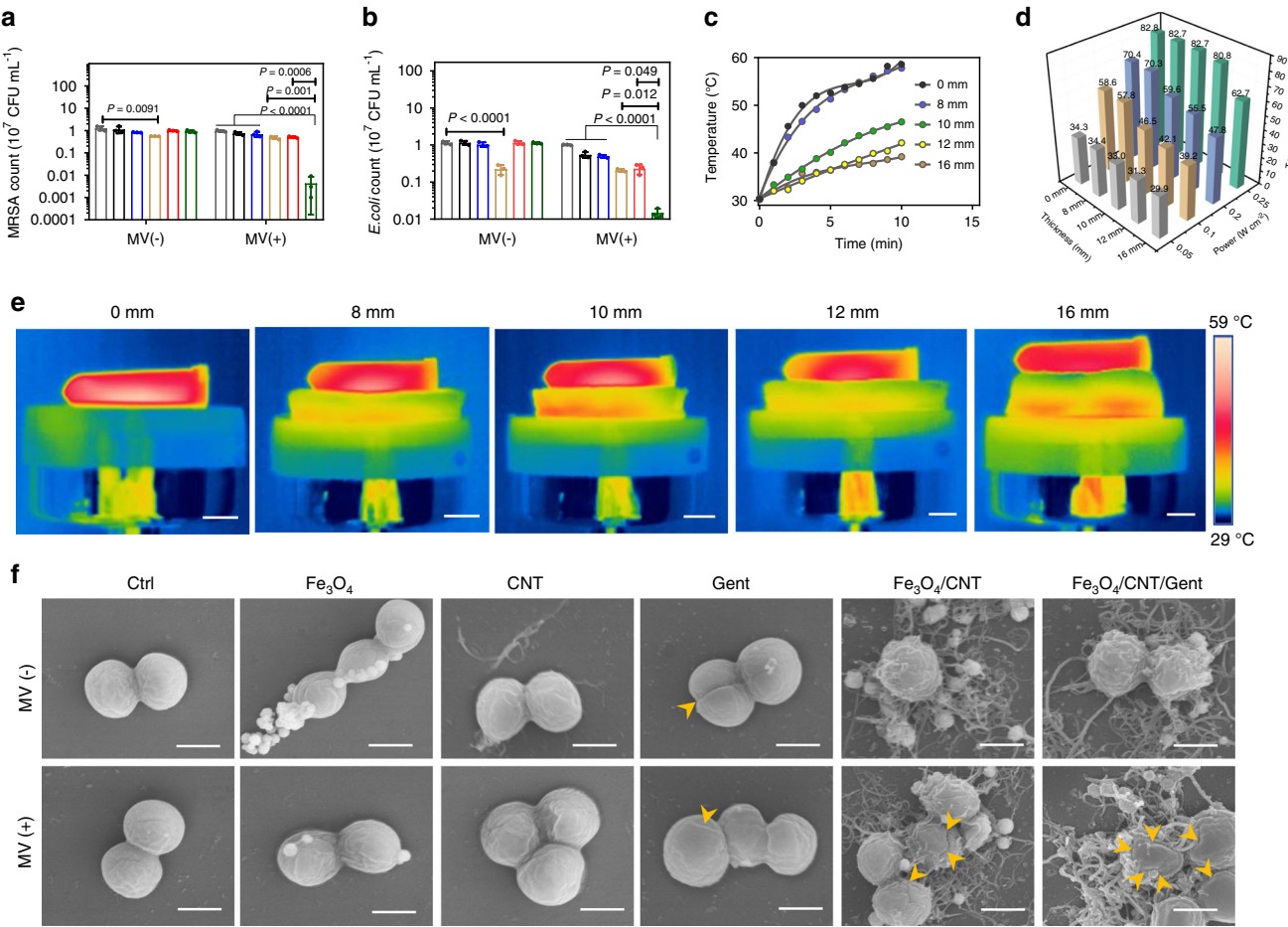

**Fig. 5 Evaluation of in vitro tissue penetration depth and antibacterial activity.** MRSA (**a**) and *E. coli* (**b**) strain counts calculated from spread-plate assays after treatment with Fe$_3$O$_4$, CNT, Fe$_3$O$_4$/CNT, or Fe$_3$O$_4$/CNT/Gent (1 mg mL$^{-1}$) under MV excitation for 20 min or not. **c** Microwaveocalortic curves of Fe$_3$O$_4$/CNT/Gent (1 mg mL$^{-1}$) at different thicknesses of pork tissue (0, 8, 10, 12, and 16 mm) under MV (0.1 W cm$^{-2}$, 10 min) excitation. **d** Temperature elevations of Fe$_3$O$_4$/CNT/Gent (1 mg mL$^{-1}$) on different thicknesses of pork tissue and MV power density via microwaveocalortic conversion. **e** Infrared thermal images of Fe$_3$O$_4$/CNT/Gent (1 mg mL$^{-1}$) through different thicknesses (0, 8, 10, 12, and 16 mm) of pork tissue under MV excitation for 10 min. **f** SEM images of MRSA morphologies before and after treatment. Scale bars, 1 cm (**e**) and 500 nm (**f**). Data are presented as mean ± standard deviations from a representative experiment (*n* = 3 independent samples). *P* values were analysed by one-way ANOVA with Tukey's multiple comparisons post hoc test. Grey circles indicate the group of Ctrl, black circles indicate the group of Fe$_3$O$_4$, blue circles indicate the group of CNT, brown circles indicate the group of Gent, red circles indicate the group of Fe$_3$O$_4$/CNT, green circles indicate the group of Fe$_3$O$_4$/CNT/Gent. Source data are provided as a Source Data file.

order to characterize the affinity of Fe$_3$O$_4$/CNT/Gent for different chemical compositions, the aqueous solution of Fe$_3$O$_4$/CNT/Gent was first put into a nonpolar medium (chloroform) to form a layered solution due to the different polarities of the water and chloroform. Subsequently, equal volume of oleylamine (C$_{17}$H$_{33}$–NH$_2$), oleic acid (C$_{17}$H$_{33}$–COOH), or oleyl alcohol (C$_{17}$H$_{33}$–OH), was added respectively. When Fe$_3$O$_4$/CNT/Gent exhibits a higher affinity to chemical groups, it will induce more Fe$_3$O$_4$/CNT/Gent transfer from the upper aqueous solution to the lower chloroform solution, thus producing a corresponding clearer upper solution. As shown in Supplementary Fig. 16, after adding oleylamine (C$_{17}$H$_{33}$–NH$_2$), it exhibited the clearest upper aqueous media, followed by oleyl alcohol (C$_{17}$H$_{33}$–OH) and finally oleic acid (C$_{17}$H$_{33}$–COOH), suggesting that oleylamine (–NH$_2$) had the best affinity for Fe$_3$O$_4$/CNT/Gent. Meanwhile, the nearly completely transparent and clear upper solution produced by adding oleylamine indicated that Fe$_3$O$_4$/CNT/Gent was drawn almost completely from the upper aqueous solution to the bottom chloroform. These results suggested that the bacteria-capturing activity of the Fe$_3$O$_4$/CNT/Gent originated from the high and selective binding affinity of Fe$_3$O$_4$/CNT/Gent to the

amino groups on the surface of the bacteria, which was strongly accordant with the observation (Fig. 4b) that Fe$_3$O$_4$/CNT/Gent homogeneously covered the surface of MRSA with a thick outer layer of peptidoglycan consisting of the disaccharide and amino acids[36]. Similarly, the preferential capture of Fe$_3$O$_4$/CNT/Gent at the two poles of *E. coli*, which has an outer membrane of lipopolysaccharides consisting of lipids and polysaccharides, may be owing to the enriched amino groups of proteins exposed at the poles of *E. coli*[37].

**In vitro tissue penetration depth and antibacterial activity.** First, we observed the viability of MRSA and *E. coli* in the absence of pork tissue before and after MV treatment. For the control (Ctrl, physiological saline) groups without and with MV treatment, there were no significant differences in either the MRSA or *E. coli* counts, as shown in Fig. 5a and b. In the absence of MV excitation (MV−), the MRSA count (10$^7$ CFU mL$^{-1}$) was inconspicuously reduced in the Fe$_3$O$_4$, Fe$_3$O$_4$/CNT, and Fe$_3$O$_4$/CNT/Gent groups, respectively (Fig. 5a); a similar phenomenon was observed for *E. coli* (Fig. 5b). The Gent group showed an

obvious reduction of both MRSA and *E. coli* counts, with anti-bacterial rates of $56.177 \pm 0.588\%$ and $80.517 \pm 5.933\%$, respectively, indicating that Gent is more sensitive to *E. coli* than MRSA. Under MV irradiation (MV+), both MRSA and *E. coli* counts ($10^7$ CFU mL$^{-1}$) of the Fe$_3$O$_4$/CNT/Gent group fell to 0.004 ($P < 0.0001$, compared to Ctrl, Fe$_3$O$_4$, and CNT; $P = 0.001$, compared to Gent; and $P = 0.0006$, compared to Fe$_3$O$_4$/CNT) and 0.015 ($P < 0.0001$, compared to Ctrl, Fe$_3$O$_4$, and CNT; $P = 0.012$, compared to Gent; and $P = 0.049$, compared to Fe$_3$O$_4$/CNT, respectively), far lower than other groups, demonstrating that $99.556 \pm 0.427\%$ MRSA and $98.529 \pm 0.404\%$ *E. coli* were killed by the synthesized nanocapturers during 20 min of MV treatment. These results were confirmed by the spread-plate assay (Supplementary Fig. 17). In addition, the antibacterial rates of Fe$_3$O$_4$/CNT/Gent for MRSA after heating for 5, 10, 15, and 20 min were 6.96%, 47.82%, 72.68%, and 99.72%, respectively (Supplementary Fig. 18). Thus, 20 min MV irradiation was chosen for in vivo experiment. Notably, the temperature of Fe$_3$O$_4$/CNT/Gent was higher than 50 °C after MV irradiation for 5 min and then maintained below 55 °C for another 15 min. During this course, the temperature is adjustable by controlling the MV power, so that minimal invasion of normal tissues and highly effective bacteria-killing can be achieved in a short time[12]. The synergistic effect of Gent and MCT has also been verified and is expressed by the combination index[38], where the combined value is <1, indicating a strong synergy between Gent and MCT (Supplementary Table 2).

Next, we investigated the tissue penetration depth of the Fe$_3$O$_4$/CNT/Gent for microwaveocaloric conversion and anti-MRSA behaviors under MV irradiation. As shown in Fig. 5c, after MV excitation (0.1 W cm$^{-2}$, 10 min), the temperature of Fe$_3$O$_4$/CNT/Gent solution decreased as the pork tissue thickness was increased when it reached over 10 mm. Especially, to observe the antibacterial activities of Fe$_3$O$_4$/CNT/Gent solution with different pork thickness, fluorescence staining was carried out (Supplementary Fig. 19), and the antibacterial efficacy of Fe$_3$O$_4$/CNT/Gent was calculated to be 100%, 98.55%, and 86.99%, respectively under the pork thickness of 0, 8, and 10 mm after the MV (0.1 W cm$^{-2}$, 20 min) irradiation. In contrast, under the thickness of 12 and 16 mm, the antibacterial rate was <10% under the same MV irradiation, which was ascribed to the temperature drop as the thickness of the penetrating tissue increased. The measured temperature varied with the tissue depths and the MV power (Fig. 5d), suggesting that the differing depth penetrations were achieved by adjusting MV power. The corresponding infrared thermal images visually disclosed that the efficient heating of the Fe$_3$O$_4$/CNT/Gent solution was achieved without causing significant heating inside the tissues (Fig. 5e).

The bacterial morphologies from the different groups were observed using SEM (Fig. 5f and Supplementary Fig. 20). In the case of the control group, with or without MV irradiation, the images showed the integrated membranes with smooth surfaces and the typical spherical MRSA morphology. The same intact morphology was also observed in the MRSA samples from the Fe$_3$O$_4$, Fe$_3$O$_4$/CNT, and Fe$_3$O$_4$/CNT/Gent without MV irradiation groups, although the CNT and Gent groups showed slightly distorted membranes. In contrast, when exposed to MV, broken and distorted MRSA membranes (indicated by orange arrows) were observed in the Fe$_3$O$_4$/CNT and Fe$_3$O$_4$/CNT/Gent groups. Partially distorted bacteria were found in the CNT and Gent groups, which were almost intact in Ctrl and Fe$_3$O$_4$ groups. Similar results were also observed in *E. coli* (Supplementary Fig. 20). These above results confirmed that single factors such as Gent or microwaveocaloric action exhibit lower antibacterial efficacy, but the synergistic action of both the microwaveocaloric effect and Gent can exhibit far higher antibacterial efficacy than either can alone.

Accordingly, we propose that the mechanism underlying the broad-spectrum bactericidal action of Fe$_3$O$_4$/CNT/Gent nanocapturer is as follows. First, Fe$_3$O$_4$/CNT/Gent precisely captures the bacteria by anchoring with the amino groups of the bacterial surface, favoring in situ treatment and thus improving the therapeutic efficacy of MCCT. Then, the microwaveocaloric effect of Fe$_3$O$_4$/CNT/Gent effectively damages the bacterial membrane and releases the Gent simultaneously. Meanwhile, the destruction of the bacterial membrane reduced its heat resistance, allowing the further permeation of released Gent (Fig. 1). This synergistic antibacterial behavior of Fe$_3$O$_4$/CNT/Gent will improve our knowledge the antibacterial applications of MV-responsive materials.

**In vivo eradication of osteomyelitis.** Because Fe$_3$O$_4$/CNT/Gent nanocapturers demonstrated strong efficacy of MCCT in vitro, as well as excellent biosafety (Supplementary Fig. 21) and superior blood circulation (Supplementary Fig. 22), they were investigated for further applications in a rabbit model of osteomyelitis. As shown in Fig. 6a, the macroscopic imaging exhibited apparent festering/swelling (indicated by white dashed circles and magnified) in the Ctrl (osteomyelitis treatment with physiological saline) and MV (osteomyelitis treatment with MV only) groups, which indicated a severe bacterial infection. In contrast, the femur and tibia specimens in the Fe$_3$O$_4$/CNT/Gent + MV (osteomyelitis treatment with Fe$_3$O$_4$/CNT/Gent and MV) and Fe$_3$O$_4$/CNT/Gent + MV + MF (osteomyelitis treatment with Fe$_3$O$_4$/CNT/Gent, MV, and applied magnetic field) groups were normal without apparent swelling or pustules, demonstrating no infection. Meanwhile, as shown in the Wright-stained (Fig. 6b) samples, the numbers of bacteria (indicated by green arrows) and lymphocytes (indicated by yellow arrows) decreased following Fe$_3$O$_4$/CNT/Gent-targeted treatment, especially in the Fe$_3$O$_4$/CNT/Gent + MV + MF group. In contrast, the obvious bacteria and lymphocytes were observed in bone marrow treated with the Ctrl group. To quantify the antibacterial properties of Fe$_3$O$_4$/CNT/Gent, we performed the colony-count assay using the harvested bone marrow tissues at 2 (Fig. 6c) and 14 days (Supplementary Fig. 23). All the treatment groups showed a reduction of bacterial colonies in the MRSA-infected bone marrow. Of these, the Fe$_3$O$_4$/CNT/Gent + MV + MF group always demonstrated the greatest antibacterial efficacy ($P = 0.0348$ compared to the Fe$_3$O$_4$/CNT/Gent + MV group at 2 days) against MRSA in vivo.

Magnetic resonance imaging (MRI) and hematoxylin and eosin (H&E) staining were used to study the therapeutic effects of Fe$_3$O$_4$/CNT/Gent for MRSA-induced osteomyelitis and its damage to tissues. According to 3D-MRI (Fig. 6d), we observed an obvious separation of bone marrow and bone tissue in the Ctrl group, and the blurred signal was clearly seen in the bone marrow in the MV group, these demonstrating that bone marrow was infected and partially damaged, thus causing it to separate from the bone tissues. Similarly, H&E staining at 2 and 7 days showed visible lymphocytes (indicated by orange arrows), multinucleated giant cells (indicated by green arrows), and neutrophil (indicated by red arrows) in MRSA-infected bone marrow (Fig. 6e). Many inflammatory cells appeared because of severe inflammation and bacterial infection. After 14 days of treatment, some inflammatory cells remained in the bone marrow of both the Ctrl and MV groups and the structure of fat cells was destroyed accompanied by myelofibrosis. In contrast, clear structured tissues were observed in both Fe$_3$O$_4$/CNT/Gent + MV and Fe$_3$O$_4$/CNT/Gent + MV + MF groups (Fig. 6d), indicating that the MRSA infecting bone marrow had been almost eradicated. Less inflammation and destruction also were present in both the Fe$_3$O$_4$/CNT/Gent + MV and Fe$_3$O$_4$/CNT/Gent + MV + MF

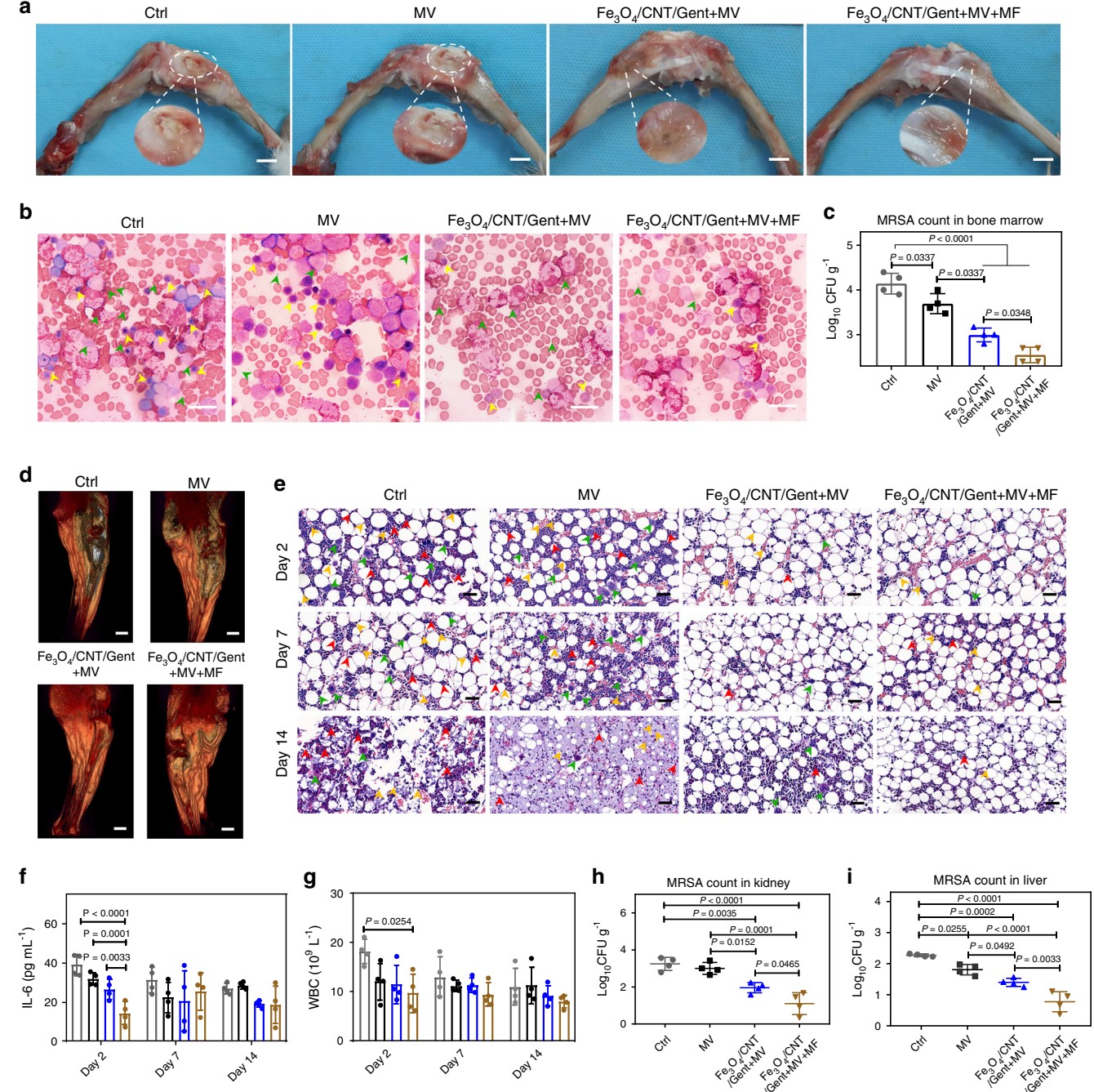

**Fig. 6 Antibacterial effects of Fe₃O₄/CNT/Gent on MRSA-infected osteomyelitis in vivo. a** Macroscopic image of the infected femur and tibia specimens of rabbit models after 14 days of treatment (the festering tissues are indicated by white dashed circles and magnified). Scale bars = 1 cm. **b** Wright-stained images of infected bone marrow tissue after 14 days of treatment. Scale bars = 20 μm. **c** The MRSA counts in the infected bone marrow after 2 days with different treatments. **d** 3D-MRI after 14 days treatment. Scale bars = 1 cm. **e** H&E staining images of infected bone marrow tissues after 2, 7, and 14 days of treatment. Scale bars = 50 μm. IL-6 levels (**f**) and amount of WBC (**g**) from 2 to 14 days in blood. At 14 days post-treatment, samples were collected from harvested kidneys (**h**) and livers (**i**) for bacterial quantification. Data are presented as mean ± standard deviations from a representative experiment (n = 4 biologically independent samples). *P* values were analysed by one-way ANOVA with Tukey's multiple comparisons post hoc test. Grey circles indicate the group of Ctrl, black circles indicate the group of MV, blue circles indicate the group of Fe₃O₄/CNT/Gent + MV, brown circles indicate the group of Fe₃O₄/CNT/Gent + MV + MF. Source data are provided as a Source Data file.

groups (Fig. 6d), suggesting that Fe₃O₄/CNT/Gent is highly effective at killing MRSA in vivo with MCCT assistance.

We collected and analyzed the blood samples of the rabbits after 2, 7, and 14 days of treatment. The concentrations of interleukin-6 (IL-6, Fig. 6f), sensitive factors for diagnosing bacterial infections, and white blood cells (WBC) counts (Fig. 6g)

were tested to evaluate the inflammatory response. After 2 days of surgery, the IL-6 (*P* < 0.0001) and WBC (*P* = 0.0254) levels of the Fe₃O₄/CNT/Gent + MV + MF group were significantly lower than that the Ctrl group, indicating that the bacterial infection was restrained in the Fe₃O₄/CNT/Gent + MV + MF group due to the dual-targeting effect of the precise MRSA-capturing and the

magnetic field. The organ injuries of the infected rabbits were investigated in vivo. At 14 days post-treatment, the important metabolic organs (kidney and liver) were harvested to quantify the MRSA present. As shown in Fig. 6h, i, the $Fe_3O_4$/CNT/Gent + MV and $Fe_3O_4$/CNT/Gent + MV + MF groups both showed a reduced bacterial burden in the kidneys and livers of the infected rabbits and that the latter was more effective than the former ($P = 0.0465$ for kidney; $P = 0.0033$ for liver), suggesting that the magnetic field enhanced the treatment effect of the nanocapturer by inhibiting the systemic spread of the MRSA. Rabbits with severe bacterial infection are usually accompanied by organ injury. As shown in Supplementary Fig. 24, the Ctrl- and MV-treated rabbits manifested vasodilatation congestion (indicated by green arrows) and punctate necrosis of cardiomyocytes (indicated by yellow arrows) in the heart, accompanied by protein mucus exudates in the myelin of spleen (indicated by orange arrows), and many inflamed cells (indicated by blue arrows) appeared in the liver, lung, and spleen. In contrast, treatment with $Fe_3O_4$/CNT/Gent + MV, these abnormalities related to MRSA-induced organ injury was partially alleviated, especially for $Fe_3O_4$/CNT/Gent + MV + MF, the best therapeutic effects had been achieved. These findings revealed that $Fe_3O_4$/CNT/Gent + MV + MF effectively reduced the harm of osteomyelitis in rabbits by reducing the number of bacteria and alleviating organ injury.

## Discussion

A precise and synergistic antibacterial strategy integrating the advantages of dual targeting (material targeted binding and magnetic targeting) and MCCT is proposed. We successfully synthesized the MV-responsive $Fe_3O_4$/CNT/Gent, which effectively eliminated the MRSA infection in deep tissues. The mechanism of enhanced microwaveocaloric conversion efficiency of $Fe_3O_4$/CNT/Gent is ascribed to high MV absorption performance (~88.81% absorption) arising from good impedance matching, multiple scattering, interfacial polarization, dipole polarization, conductive loss, and natural ferromagnetic resonance loss. Simultaneously, CNT-endowed $Fe_3O_4$/CNT/Gent with excellent bacteria-capturing capabilities and the different capturing effects of $Fe_3O_4$/CNT/Gent on MRSA and *E. coli* may be related to their unique cell membrane components. The underlying mechanism of the efficiently and precisely eradicating osteomyelitis action of $Fe_3O_4$/CNT/Gent originated from the synergistic reaction of the synthesized nanocapturer binding bacteria first then producing heat under MV stimulation, followed by Gent release. Finally, magnetic action can prevent MRSA from spreading via the blood, avoiding sepsis and organ infections. This synergistic strategy is versatile and not confined to bone marrow infection—it also exhibits great promise as an advanced MCT tool for treating a variety of bacterial infected deep diseases in clinical settings.

## Methods

**Materials for experiments**. All the starting materials were purchased from commercial suppliers. Pure multi-walled CNTs were purchased from Aiweixin Chemical Technology Co., Ltd. (Tianjin, China). $FeCl_3 \cdot 6H_2O$ and sodium acetate were purchased from Yuanye Biological Technology Co., Ltd. (Shanghai, China). The 1-tetradecanol and gentamicin used were purchased from Aladdin (Shanghai, China). Hexadecyltrimethylammonium bromide (CTAB) was purchased from Solomon Biotechnology Co., Ltd. (Tianjin, China). Sulfuric acid ($H_2SO_4$, 98%) and nitric acid ($HNO_3$, 65–68%) were purchased from Jiangtian Chemical Technology Co., Ltd. (Tianjin, China). The rabbit Interleukin-6 ELISA Kit (CSB-E06903Rb) was purchased from Huamei Biological Engineering Co., Ltd. (Wuhan, China). All chemical solvents and salts used were of analytical grade. Starting materials were used without further purification unless otherwise noted.

**Processing of pure CNT**. The functionalized CNTs powder samples were prepared in an $H_2SO_4$ and $HNO_3$ soak. Briefly, (1) The pure CNTs were soaked in $H_2SO_4$

overnight and then stirred for 72 h; (2) the residual acid was washed off the CNTs using deionized water (d$H_2O$) and the CNTs vacuum-dried; (3) the treated carbon tubes were dispersed in $H_2SO_4$ and stirred overnight under $N_2$ airflow; the same volume of 68% $HNO_3$ (diluted with d$H_2O$) was added and stirred the mixture comprising at 65 °C for 2 h; (4) the CNTs were filtered and washed with methanol; and the CNTs were vacuum-dried to obtain the functionalized CNTs.

**Synthesis of $Fe_3O_4$ and $Fe_3O_4$/CNT**. The $Fe_3O_4$/CNT was prepared as follows: $FeCl_3 \cdot 6H_2O$ (0.54 g) and CNTs (0.15 g) were dissolved in ethylene glycol (16 mL) and stirred for 30 min at room temperature. Polyethylene glycol (PEG 8000, 0.4 g) and CTAB (1.2 g) were added to the resulting solution, which was then stirred vigorously for 30 min. Finally, sodium acetate (1.44 g) was added and the solution continued to be stirred until it was homogeneous. It was then transferred to a 20 mL Teflon-lined stainless-steel autoclave and heated at 200 °C for 12 h. After natural cooling, the obtained product was collected by magnetic separation and washed with ethanol and d$H_2O$ several times. Finally, the obtained product was re-dispersed in acetone and isopropanol (3:1, v/v) and refluxed at 85 °C for 6 h to remove extra CTAB. Samples of $Fe_3O_4$/CNT with various doping levels were also synthesized by regulating the dosages of CNT (12.5, 35, 150, and 225 mg). The same preparation process was used to obtain $Fe_3O_4$ nanoparticles, except that no CNTs were added during the solvothermal process.

**Synthesis of the $Fe_3O_4$/CNT/Gent nanocapturer**. Gent (50 mg) was dissolved in d$H_2O$ (5 mL), 1-tetradecanol (25 mg) was dissolved in ethanol (1 mL), and $Fe_3O_4$/CNT (50 mg) nanocomposites were dissolved in ethanol (4 mL). These solutions were all added to a 25 mL conical flask. After 5 min of ultrasonic dispersion, the solution was agitated in a shaker for 24 h under 50 °C. Finally, the precipitate was collected by magnetic separation, washed three times with d$H_2O$, and vacuum-dried.

**Characterization of $Fe_3O_4$/CNT/Gent**. The morphology of $Fe_3O_4$/CNT and $Fe_3O_4$/CNT/Gent was observed using SEM (S4800, Japan) and TEM (JEOL JEM-2100F, Japan). Elemental mapping images were obtained using a TEM (JEOL JEM-2100F, Japan). The surface compositions, chemical structure, and crystal structures were collected with XPS (250Xi, United States), FTIR (Nicolet IS10, United States), and XRD (D8 Advanced, Germany) utilizing CuKα radiation. The magnetic properties of the prepared samples were using vibrating sample magnetometer (SK-300, Japan).

**Microwaveocalortic effect measurements**. The $Fe_3O_4$, CNT, $Fe_3O_4$/CNT, and $Fe_3O_4$/CNT/Gent in concentrations of 1 mg mL$^{-1}$ (dissolved in physiological saline) were subjected to 15 min of ultrasonic dispersion and then microwaved (2.45 GHz, Schneider Medical Equipment Co., Ltd., China) for 5 min in a 2 mL Eppendorf (EP) tube. The temperature of each solution was recorded every minute using an FLIR thermal camera (FLIR E64501, Estonia). Similarly, for the microwaveocalortic on–off curve of $Fe_3O_4$/CNT/Gent (1 mg mL$^{-1}$), after 20 min of irradiation, the cooling-stage temperature was also recorded every minute until it reached nearly room temperature for three cycles.

**Drug release of $Fe_3O_4$/CNT/Gent**. To study the controlled release characteristics of Gent in $Fe_3O_4$/CNT/Gent the $Fe_3O_4$/CNT/Gent nanocomposites (1 mg) were dispersed in a sample tube filled with 1 mL of PBS (pH = 5.5). The $Fe_3O_4$/CNT/Gent was agitated in a shaker at 37 °C and then radiated by MW (20 min, 0.1 W cm$^{-1}$) at preset periods. The concentration of Gent in the supernatant of nanocomposites was determined using a UV–vis spectrophotometer (333 nm)[39]. The release of Gent at different time points was calculated from the standard curve. The standard curve was calculated as Supplementary Fig. 25:

$$Y = 0.01476X - 0.00134 \left( R^2 = 0.999 \right), \tag{2}$$

where, $Y$ represents the absorbance of Gent at 333 nm and $X$ represents the corresponding calculated Gent concentration (μg mL$^{-1}$). The cumulative release was defined as mass of released Gent/mass of loaded Gent × 100%.

**MV absorption test**. The electromagnetic parameters of $Fe_3O_4$, CNT, $Fe_3O_4$/CNT, and $Fe_3O_4$/CNT/Gent were prepared by uniformly mixing the absorbents with paraffin matrix according to a same mass fraction of 36% and compacting it into a columnar ring of with a 7.00-mm outer diameter and a 3.04-mm inner diameter. This was measured using a vector network analyzer (Agilent PNA-N5244A, USA). The MV absorption performances of $Fe_3O_4$, CNT, $Fe_3O_4$/CNT, and $Fe_3O_4$/CNT/Gent were evaluated by calculating the RL based on the transmission line theory[40].

$$z_{in} = z_0 \sqrt{\frac{\mu_r}{\varepsilon_r}} tanh \left[ j \left( \frac{2f\pi t}{c} \right) \right] \sqrt{\mu_r \varepsilon_r}, \tag{3}$$

$$RL = 20 \log \left| \frac{z_{in} - z_0}{z_{in} + z_0} \right|, \tag{4}$$

where $Z_{in}$ is the input impedance at the absorber surface, $Z_0$ is the impedance of the

air, $f$ is the MV frequency, $t$ is the thickness of the absorber, and $c$ is the velocity of light in free space.

The attenuation constant $\alpha$ was calculated using Eq. (5):

$$\alpha = \frac{\sqrt{2}\pi f}{c} \times \sqrt{(\mu''\varepsilon'' - \mu'\varepsilon') + \sqrt{(\mu''\varepsilon'' - \mu'\varepsilon')^2 + (\mu'\varepsilon'' + \mu''\varepsilon')^2}}. \quad (5)$$

**Bacteria-capturing activity**. Overnight bacterial cultures were adjusted to $OD_{600} = 1.0{-}1.1$ using physiological saline. Eight hundred µL of overnight bacterial cultures ($OD_{600} = 1.1$) were mixed with 200 µL of physiological saline (negative control), $Fe_3O_4$ solution (5 mg mL$^{-1}$), CNT solution (5 mg mL$^{-1}$), $Fe_3O_4$/CNT solution (5 mg mL$^{-1}$), $Fe_3O_4$/CNT/Gent solution (5 mg mL$^{-1}$), and Gent solution (5 × drug loading dose), respectively. Each culture was added to a 2 mL EP tube and incubated in a shaker (150 r.p.m. with a rotational radius of 10 cm) for 20 min at room temperature. The resulting solutions were placed in optical cuvettes, which were then placed on top of or alongside a magnet (neodymium rare earth permanent magnet with grade N38 magnetic energy, around 30 mm × 20 mm × 10 mm in length, width, and height, respectively) for 4 min to collect the magnetic nanoparticles. The bacteria-capturing activity was determined by the $OD_{600}$ of the supernatants. Similarly, bacteria cells ($OD_{600} = 1.1$) were prepared and added to different concentrations of the $Fe_3O_4$/CNT/Gent solution (the final concentrations were 0, 0.25, 0.5, and 1 mg mL$^{-1}$, respectively). After being shaken for 0, 10, 20, 60, 120, and 180 min at room temperature the samples were placed on a magnet to collect the $Fe_3O_4$/CNT/Gent-captured bacteria and the supernatant was collected for OD value testing with a microplate reader (SpectraMax I3MD, USA). The same method was used to determine the MRSA-capturing effect of different CNT-doped $Fe_3O_4$/CNT hybrids: 800 µL of bacterial culture ($OD_{600} = 1.0$) was mixed with 200 µL of physiological saline (shaken for 20 min; after magnet-induced separation) as the Ctrl group; 800 µL of bacterial culture ($OD_{600} = 1.0$) was mixed with 200 µL of the hybrid (1 mg mL$^{-1}$, shaken for 20 min, after magnet-induced separation) as the experimental group. In order to directly observe the different bacteria-capturing capacity of $Fe_3O_4$/CNT/Gent (1 mg mL$^{-1}$, shaken for 20 min) to both MRSA (CCTCC 16465) and E. coli (ATCC 8099), 20 µL of the suspensions were removed before and after magnetic induced separation, appropriately diluted in a liquid culture medium, plated on the Luria–Bertani (LB) agar plates, and cultured at 37 °C for 20 h.

**Antibacterial test**. The in vitro antibacterial activity of $Fe_3O_4$, CNT, Gent, $Fe_3O_4$/CNT, and $Fe_3O_4$/CNT/Gent against MRSA and E. coli was quantitatively using the spread-plate method. After being cultured for ~24 h, the MRSA and E. coli (≈$10^9$ CFU mL$^{-1}$) were diluted to ≈$10^7$ CFU mL$^{-1}$ in LB medium for subsequent experiments. The bacterial suspensions were then incubated with physiological saline (control), $Fe_3O_4$ (1 mg mL$^{-1}$), CNT (1 mg mL$^{-1}$), Gent (7.31 µg mL$^{-1}$, drug loading dose), $Fe_3O_4$/CNT (1 mg mL$^{-1}$), and $Fe_3O_4$/CNT/Gent (1 mg mL$^{-1}$) in 2 mL EP tubes with and without MV irradiation at a power density of 0.1 W cm$^{-2}$ for 20 min. Specifically, the temperature of irradiated $Fe_3O_4$/CNT and $Fe_3O_4$/CNT/Gent (1 mg mL$^{-1}$) was more than 50 °C after 5 min and <55 °C after 15 min. After different treatments, the diluted bacterial solution (20 µL) was smeared on LB agar plate evenly and cultured at 37 °C. The colonies were counted after cultured 20 h to calculate the antibacterial ratio, which was then assessed using Eq. (6):

$$\text{Antibacterial ratio (\%)} = \frac{A - B}{A} 100\%, \quad (6)$$

where $A$ and $B$ represent the numbers of bacteria in the control group and experimental group, respectively.

The morphologies of the bacteria (both MRSA and E. coli) interacting with $Fe_3O_4$, CNT, Gent, $Fe_3O_4$/CNT, and $Fe_3O_4$/CNT/Gent were evaluated using SEM. The treated bacteria were soaked with glutaraldehyde (2.5%) solution for 4 h and washed with PBS (pH = 7.0). The samples were then dehydrated in different concentrations (30, 50, 70, 90, and 100%, v/v) of ethanol for 15 min and air-dried before observation.

**In vitro cytotoxicity evaluation**. The MC3T3-E1 (ATCC CRL-2593) and NIH-3T3 (ATCC CRL-1658) were cultured in Dulbecco's modified eagle medium and minimum essential medium alpha (α-MEM) respectively, including supplemented with 10% (v/v) fetal bovine serum, 1% amphotericin and 1% penicillin–streptomycin and incubated at 37 °C in 95% humidity and an atmosphere containing 5% $CO_2$.

For the fluorescence morphology, MC3T3-E1 cells were first cultured in six-well plates at 37 °C for 24 h and further incubated with $Fe_3O_4$, CNT, Gent, $Fe_3O_4$/CNT, and $Fe_3O_4$/CNT/Gent (concentrations are consistent with those used in antibacterial experiments). The treated cells were cultured for another 24 h. After incubation, the samples were washed with PBS, soaked in formaldehyde (4%) for 10 min, and rinsed with sterile PBS. These samples were then stained with fluorescein isothiocyanate-phalloidin for 30 min (avoid light during this process), washed with PBS, stained again with 4,6-Diamidino-2-phenylindole for 20 s, rinsed with PBS three times, and then photographed using laser scanning confocal microscopy (Nikon A1R+, Japan).

MTT assay of MC3T3-E1. The various samples ($Fe_3O_4$, CNT, Gent, $Fe_3O_4$/CNT, and $Fe_3O_4$/CNT/Gent) and various concentrations of $Fe_3O_4$/CNT/Gent were co-cultured with MC3T3-E1 cells ($10^5$ cells/well) for 3 days ($n = 5$ independent

samples). The control group was physiological saline. After incubation certain periods (3 days), the medium was removed and 3-(4,5-dimethylthiazol-2-yl)−2,5-diphenyltetrazolium bromide (MTT, 0.5 mg mL$^{-1}$) was added, this resulting solution incubation for 4 h at 37 °C. The solution was then removed and 200 µL dimethyl sulfoxide was added (shook for 15 min) and centrifuged. Finally, the absorption of the supernatant at 570 nm was determined. The cell viability (%) was calculated by comparing the absorbance values of these samples with the control.

Evaluation hemolysis of $Fe_3O_4$/CNT/Gent nanocapturer. First, 5 mL of New Zealand rabbit blood was diluted with 50 mL of PBS solution and centrifuged in the mixture solution at 176 × $g$ for 6 min to obtain red blood cells (RBCs). The RBCs were then washed three times with PBS and resuspended in 20 mL of the PBS solution. Different concentrations of $Fe_3O_4$/CNT/Gent nanocapturer were incubated with diluted RBCs for 4 h at 37 °C ($n = 3$ independent samples). After centrifugation for 6 min at 176 × $g$, the absorbance of the supernatant was measured at 405 nm. A positive control (100% lysis) was prepared by treating RBCs with 1% Triton X-100. The hemolysis percentage was calculated by comparing the absorbance values of these samples with the positive control.

**In vivo safety**. In brief, male New Zealand rabbits (6–8 weeks old, ~2.0–2.5 kg in weight) were first shaved to remove the hair on the hind legs. Subsequently, 200 µL of 5 mg mL$^{-1}$ of $Fe_3O_4$/CNT/Gent nanocapturer was injected in situ, using rabbits that had not been subjected to surgery as the control. At different time points post-injection, the complete blood was collected for blood routine tests (WBC, RBC, HGB, MCH, PLT, MPV, MCV, Gran, HCT, Mon, and RDW; 1 day, 7 days, and 14 days post-injection) using an automatic animal blood cell analyzer (BC-2800vet, China). The concentrations of Fe ions ($Fe^{2+}$ and $Fe^{3+}$) were then measured using an ICP-MS (Agilent 7800, USA). The internal organs (heart, liver, spleen, lung, and kidney) were collected 14 days post-injection for histological analysis by H&E staining.

**Rabbit osteomyelitis model and treatment**. The study was carried out in accordance with the Guide for the Care and Use of Laboratory Animals of the National Institutes of Health. The ethical aspects of the animal experiment were approved by the Animal Ethical and Welfare Committee (AEWC) of the Institute of Radiation Medicine, Chinese Academy of Medical Sciences (Approval No. IRM-DWLL-2019087). The male New Zealand white rabbits (6–8 weeks old, ~2.0–2.5 kg in weight) were randomly divided into four groups ($n = 4$ per group) at every preset time point: the control group (physiological saline), the MV treatment group, the $Fe_3O_4$/CNT/Gent + MV treatment group, and the dual-targeting ($Fe_3O_4$/CNT/Gent + MV + MF) group.

The rabbits were under a 25 ± 2 °C and 60–70% (humidity) conditions for 3 days. They were anaesthetized with an intramuscular injection of pentobarbital (30 mg kg$^{-1}$) prior to surgery. After anesthesia, the hind legs of the rabbits were shaved and disinfected. A hole was drilled in the tibial plateau of the hind legs using medical electric drill 2 mm in diameter; via this hole, the legs were injected with $10^6$ CFU (50 µL) of MRSA bacterial suspension in situ to establish the osteomyelitis model. Physiological saline or $Fe_3O_4$/CNT/Gent was added for different treatment options. Finally, the holes were sealed with bone wax and the muscles and skin were sutured carefully. The rabbits were subjected to different treatments: The control group was injected with physiological saline (200 µL). The MV group was injected with physiological saline (200 µL), and then irradiated MV for 20 min (2.45 GHz, 0.1 W cm$^{-2}$, keeping the temperature between 50 and 55 °C once a temperature of 55 °C was reached); and the $Fe_3O_4$/CNT/Gent + MV group was injected with 200 µL of $Fe_3O_4$/CNT/Gent (5 mg mL$^{-1}$) and irradiated with MV for 20 min (2.45 GHz, 0.1 W cm$^{-2}$, keeping the temperature between 50 and 55 °C once a temperature of 55 °C was reached). The dual-targeting group was treated simply by adding a strong magnet (neodymium rare earth permanent magnet with grade N38 magnetic energy, around 20 mm × 10 mm × 10 mm in length, width, and height, respectively) to the $Fe_3O_4$/CNT/Gent + MV group experiment.

To evaluate the therapeutic effect of different treatments for MRSA-infected osteomyelitis, the infected femur and tibia specimens of the rabbit models were observed, photographed, and was investigated using 3.0 T MR scanner (GE, MR750, USA) after 14 days of treatment. At a preset time, the rabbits were sacrificed, their blood was collected for blood tests (IL-6 and WBC tests), their bone marrow was collected for Wright's staining (Days 14) and H&E staining (Days 2, 7, and 14).

After 14 days of treatment, the heart, liver, spleen, lung, and kidney were stained with H&E. The bone marrow and the rest of the spleen and kidney tissues were weighed separately and quantitative of LB medium was added for every 1 g of tissue and homogenization. Homogenates (20 µL), properly diluted, were smeared on LB agar plate evenly. The number of bacteria were counted after incubation 20 h at 37 °C and was expressed as 1 g CFU g$^{-1}$ for tissues.

**Statistics and reproducibility**. All the quantitative data in each experiment were evaluated and analysed by one-way or two-way analysis of variance and expressed as the mean values ± standard deviations from at least three independent experiments, followed by Tukey's multiple comparisons post hoc test to evaluate the statistical significance of the variance. The n.s. present $P > 0.05$ and ****$P < 0.0001$ were considered statistically significant.

**Reporting summary**. Further information on research design is available in the Nature Research Reporting Summary linked to this article.

## Data availability
Data are available from the corresponding author upon reasonable request. Source data are provided with this paper.

## Code availability
Custom-written MATLAB scripts are available from the corresponding author upon reasonable request.

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

## Acknowledgements
This work is jointly supported by the National Science Fund for Distinguished Young Scholars 51925104, National Natural Science Foundation of China nos. 51871162, 51671081, and 81870809, NSFC key program 51631007, Natural Science Fund of Hubei Province, 2018CFA064, RGC/NSFC (N_HKU725-1616), Hong Kong ITC (ITS/287/17, GHX/002/14SZ), as well as Health and Medical Research Fund (No. 03142446).

## Author contributions
Y.Q., X.L., and S.W. conceived and designed the concept of the experiments. Y.Q. performed the experiments and conducted the material characterizations. Y.Q., X.L. and S.W. analyzed the experimental data and co-wrote the paper. X.L., B.L., Y.H., Y.Z., K.W.K.Y., C.L., Z.C., Y.L., Z.L., S.Z., X.W., and S.W. provided important experimental insights and performed data analysis partially. All the authors discussed, commented and agree on the paper.

## Competing interests
The authors declare no competing interests.
