## [Peer Review File · Nature Communications]

Reviewers' Comments:

Reviewer #1:

Remarks to the Author:

he manuscript by Qiao et al. reported a microwave-excited antibacterial nanocapturer for deep tissue infections that consists of microwave-responsive Fe₃O₄/CNT and the chemotherapy agent gentamicin (Gent). This system has been proven to efficiently target and eradicate methicillin-resistant *Staphylococcus aureus*-infected rabbit tibia osteomyelitis. Its robust antibacterial effectiveness was attributed to the precise bacteria-capturing ability and magnetic targeting of the nanocapturer, as well as the subsequent synergistic effects of precise microwavecaloric therapy from Fe₃O₄/CNT and chemotherapy from the effective release of antibiotics in infection sites. I would like to recommend the publication of this work in Nature communications after the following comments are addressed:

1. For the Fig. 2d, the exact FT-IR peak positions should be plotted correctly. Where are the peaks at "1110 cm⁻¹"?
2. It is not clear whether Gent molecules were encapsulated into Fe₃O₄/CNT. The authors are advised to provide more characterization data to verify the loading amount for Gent.
3. From the TEM images, it can be seen that Fe₃O₄/CNT/Gent nanocapter seems to be in the state of reunion. Authors should provide the stability data of Fe₃O₄/CNT/Gent nanocapter, especially under the complex physiological conditions.
4. Since the synergistic anti-infection effect is an important aspect of the manuscript, more quantitative results (for example, the calculation of the synergy index value) are needed to further verify it. Please use experiments to verify.
5. Page 8 Line 137: It is not clear why the zeta potential value of the materials could validate the successful loading of Gent and PCM in Fe₃O₄/CNT?
6. Fe₃O₄ or CNT alone has no microwavecalortic effect, but Fe₃O₄/CNT has microwavecalortic effect. This should be addressed or discussed in depth, and the heat conversion efficiency of the microwave-excited Fe₃O₄/CNT needs to be provided.
7. Page 9 Line 162-168, the author points out that Fe₃O₄/CNT/Gent has degradation characteristics, but it can be observed from the SEM of supplementary Fig. 10 that only the morphology of Fe₃O₄/CNT/Gent has changed. Degradation is not Disintegration? It's a change in shape?
8. In the manuscript, the author only gives the characterization data of microwave-excitation Fe₃O₄/CNT/Gent release Gent, and should also provide the Gent release curve of Fe₃O₄/CNT/Gent without microwave-excitation.
9. Supplementary Fig. 11 showed that Fe₃O₄/CNT/Gent have excellent heating effect in 5 min under microwave-excitation. Is it not too long to choose 20 min in vivo experiment?
10. Word space missing throughout the article. Kindly check.
11. There are some grammatical errors in this paper, so it is suggested to be polished by a native speaker before being published.
12. References are incomplete.

Reviewer #2:

Remarks to the Author:

In the manuscript titled "A microwave-excited nanocapturer with magnetic targeting to precisely eradicate MRSA-infected osteomyelitis", Qiao et al. designed microwave-responsive system composed of Gent-loaded Fe₃O₄/CNT, which can be employed as intelligent antibacterial nanocapturer and microwave-responsive bacteria-killing agent. Their results show that this system can achieve not only precise bacteria-targeting but also magnetic targeting, thus resulting in high MRSA-binding over 95% and restricting the spread of bacteria through blood. This is very important for clinical application. Their further antibacterial experiments disclosed that this synthesized nanocapturer possessed highly effective bacteria-killing efficacy against MRSA and

Escherichia coli within 20 min microwave irradiation, which was ascribed to the synergistic action of microwaveocaloric and chemo therapy.

Totally, this is a quite interesting paper, which is scientifically organized. The experimental design is very reasonable and feasible. The large amount of experimental data are sufficiently support their findings. I believe that this work will bring a new insight for developing novel microwave-responsive materials for rapid and effective treatment of bacteria (including MRSA and E. coli) infected diseases in deep tissues. Therefore, I enthusiastically recommend publication of this paper in Nature Communications after addressing the following minor issues:

- 1、 The author clearly analyzes the reasons why Fe₃O₄/CNT/Gent has different capture capabilities for Gram-negative bacteria and Gram-positive bacteria. Have you considered the effect of electrostatic adsorption on the capture effect? It is recommended that the author analyze this.
- 2、 The author used different thicknesses of pork (0, 8, 10, 12, and 16 mm) to prove the strong microwave penetration and recorded the corresponding temperature, but there is no calculation about the mortality rate of bacteria, it is recommended to add bacterial live death staining under different thicknesses of pork, in order to evaluate the deep sterilization effect of microwave.
- 3、 It is very important to systematically evaluate the basic cytotoxicity of the used antibacterial agent Fe₃O₄/CNT/Gent. In this manuscript, only MC 3T3-E1 osteoblast was used for evaluation. It is recommended to supplement its toxicity to connective tissues such as fibroblasts.
- 4、 The microwaveocaloric process includes the interaction between activated carrier/dipoles and microwave field. Some works devoted to analyzing polarization field and carrier motion behavior, such as Adv. Energy Mater. 2019, 9, 1902174 and Adv. Energy Mater. 2020, 1, 1904072, can help the author to thoroughly describe the related mechanism.

Response to Reviewer 1#

Original Comment: The manuscript by Qiao *et al.* reported a microwave-excited antibacterial nanocapturer for deep tissue infections that consists of microwave-responsive Fe₃O₄/CNT and the chemotherapy agent gentamicin (Gent). This system has been proven to efficiently target and eradicate methicillin-resistant *Staphylococcus aureus*-infected rabbit tibia osteomyelitis. Its robust antibacterial effectiveness was attributed to the precise bacteria-capturing ability and magnetic targeting of the nanocapturer, as well as the subsequent synergistic effects of precise microwaveocaloric therapy from Fe₃O₄/CNT and chemotherapy from the effective release of antibiotics in infection sites. I would like to recommend the publication of this work in *Nature communications* after the following comments are addressed:

Reply: We express our sincere thanks to the reviewer for his/her very positive comments and valuable suggestions, which will definitely help us further improve the quality of this work.

Comment 1: For the Fig. 2d, the exact FT-IR peak positions should be plotted correctly. Where are the peaks at “1110 cm⁻¹”?

Reply: Thank you very much for the professional suggestion. According to your suggestion, we have redrawn the accurate FT-IR peak position and marked it in the new Fig.2d. Because the FT-IR data are not normalized in the original Fig. 2d, the characteristic peak at 1110 cm⁻¹ is not easy to be observed. In our revised paper, following your advice, FT-IR curves were normalized, and the peak at 1110 cm⁻¹ can be clearly observed from the curve of Fe₃O₄/CNT/Gent in the new Fig. 2d, which was marked with an arrow as the following:

Fig. 2d FTIR spectra of the Gent, PCM, Fe₃O₄/CNT, and Fe₃O₄/CNT/Gent.

To better express this point, we have modified the manuscript as following:

In Page 8, we marked the FTIR spectra of Gent, PCM, Fe₃O₄/CNT, and Fe₃O₄/CNT/Gent as **Fig. 2d**.

Comment 2: It is not clear whether Gent molecules were encapsulated into Fe₃O₄/CNT. The authors are advised to provide more characterization data to verify the loading amount for Gent.

Reply: Thank you very much for the professional suggestion. Following your advice, except for FT-IR characterization shown in comment 1 (the FT-IR curves of Fe₃O₄/CNT/Gent, Fe₃O₄/CNT and Gent confirmed that the Gent molecules were successfully encapsulated into Fe₃O₄/CNT), in the revised paper, we have supplemented the UV-vis absorption spectra of free Gent, Fe₃O₄/CNT, and Fe₃O₄/CNT/Gent in **Supplementary Fig. 6**. Compared with the UV-vis absorption spectra of Fe₃O₄/CNT, Fe₃O₄/CNT/Gent shows a significant Gent absorption peak near 333 nm (*Adv. Funct. Mater.* 2019, 29, 1807915; *J. Pharm. Pharmacol.* 1994, 46, 718-724), which confirmed the successful loading of Gent in the Fe₃O₄/CNT.

Supplementary Fig. 6. UV-vis absorption spectra of free Gent, Fe₃O₄/CNT, and Fe₃O₄/CNT/Gent.

The loading amount of Gent was further measured by the UV-vis spectrophotometer (333 nm). According to the standard curve shown in **Supplementary Fig. 25**, the loading efficiency of Gent is 8.89 %.

Supplementary Fig. 25 The standard curve of Gent.

To better express this point, we have modified the manuscript, as followed:

In Page S7, we added the UV-vis absorption spectra of free Gent, Fe₃O₄/CNT, and Fe₃O₄/CNT/Gent as **Supplementary Fig. 6**.

In Page S30, we added the standard curve of Gent as **Supplementary Fig. 25**.

In Page 7, we added the statement “... UV-vis absorbance spectra (**Supplementary Fig. 6**) further confirmed the successful loading of Gent in the Fe₃O₄/CNT.....”

In Page 10, we added the statement “... The drug loading efficiency of Fe₃O₄/CNT/Gent is 8.89%...”

In Page 29, we added “...The release amount of Gent at different time points was calculated from the standard curve. The standard curve was calculated as **Supplementary Fig. 25**:

$$Y = 0.01476 X - 0.00134 \quad (R^2 = 0.999) \quad (2)$$

Where, y represents the absorbance of Gent at 333 nm and X represents the corresponding calculated Gent concentration (µg mL⁻¹)....”

Comment 3: From the TEM images, it can be seen that Fe₃O₄/CNT/Gent nanocapturer seems to be in the state of reunion. Authors should provide the stability data of Fe₃O₄/CNT/Gent nanocapturer, especially under the complex physiological conditions.

Reply: Thank you very much for the professional suggestion. Indeed, we agree with the reviewer that the stability of the material under the complex physiological conditions plays a vital role in clinical application.

Simulated body fluid (SBF) with ion concentrations nearly equal to those of human blood plasma is

often used to simulate complex physiological conditions *in vivo* (*Biomaterials* 2006, 27, 2907–2915). The zeta potential data is closely related with the colloid stability (*J. Control. Release.* 2016, 235, 337-351). Guidelines classifying nanoparticle-dispersions with zeta potential values of $\pm 0 - 10$ mV, $\pm 10 - 20$ mV and $\pm 20 - 30$ mV and $> \pm 30$ mV as highly unstable, relatively stable, moderately stable and highly stable, respectively are common in drug delivery literature (*J. Adv. Pharm. Technol. Res.* 2011 2, 81–87.). The zeta potential of Fe₃O₄/CNT/Gent in SBF is +23.19 mV, which proves that Fe₃O₄/CNT/Gent is relatively stable under physiological conditions. In addition, we placed Fe₃O₄/CNT/Gent in fetal bovine serum for one hour and this mixture did not precipitate, further proving the stability of Fe₃O₄/CNT/Gent nanocapturer under complex physiological conditions.

Supplementary Fig. 8 Images of Fe₃O₄/CNT/Gent (100 ppm) in fetal bovine serum setting with different times.

To better express this point, we have modified the manuscript, as followed:

In page S9, we added the images of Fe₃O₄/CNT/Gent (100 ppm) in fetal bovine serum setting with different times as **Supplementary Figure 8**.

In page 8, we made a revision: “... The Fe₃O₄/CNT/Gent in fetal bovine serum did not precipitate after setting one hour (**Supplementary Figure 8**), which proved that this hybrid is relatively stable under complex physiological conditions....”

Comment 4: Since the synergistic anti-infection effect is an important aspect of the manuscript, more quantitative results (for example, the calculation of the synergy index value) are needed to further verify it. Please use experiments to verify.

Reply: Thank you very much for the professional suggestion. To demonstrate the synergistic anti-infection effect, we calculated the synergy index value (combination index) between Gent and microwavecaloric therapy (MCT) of Fe₃O₄/CNT for eradicating MRSA:

Combination index (CI) < 1 indicates synergistic effect, CI = 1 indicates additive effect and CI > 1

indicates antagonistic effect. CI was calculated using formula based on Chou-Talalay combination index method according to the literature of *Pharmacol Rev.* **2006, 58,621–681**:

$$CI=(D)_1/ (D_x)_1 + (D)_2/(D_x)_2$$

where $(D_x)_1$ and $(D_x)_2$ represent doses of Gent and MCT in combination which are required to achieve the same efficacy as that of $(D)_1$ and $(D)_2$ represent as Gent and MCT when used alone, respectively.

$$D_x = D_m [f_a / (1-f_a)]^{1/m}$$

The parameters m , D_m , and r are the slope, antilog of the x -intercept, and the linear correlation coefficient of the median-effect plot, which signifies the shape of the dose-effect curve, and the conformity of the data to the mass-action law, respectively. D_m and m values are used for calculating the CI values.

The calculation results are shown in **Supplementary Table 2**:

Supplementary Table 2. Combination index of Gent combination for MCT resistant MRSA.

Drug	f_a	Parameter			CI
		m	D_m	r	
Gent		0.49430	63.9838	0.98002	
MCT		1.77397	1.15001	0.99691	
Gent+MCT	99.556%				0.00729

The parameters m , D_m , and r are the slope, antilog of the x -intercept, and the linear correlation coefficient of the median-effect plot, which signifies the shape of the dose-effect curve, and the conformity of the data to the mass-action law, respectively.

The CI value of Gent and MCT resistant MRSA is 0.00729, less than 1, indicating a strong synergy between Gent and MCT.

To better express this point, we have modified the manuscript, as followed:

In Page S34, we added the combination index of Gent combination for MCT resistant MRSA as **Supplementary Table 2**.

In page 19, we added: “... The synergistic effect of Gent and MCT has also been verified and is expressed by the combination index (CI)³⁸, where the combined value is less than 1, indicating a strong synergy between Gent and MCT (Supplementary Table 2)....”

Comment 5: Page 8 Line 137: It is not clear why the zeta potential value of the materials could validate the successful loading of Gent and PCM in Fe₃O₄/CNT?

Reply: Thank you very much for the professional suggestion. Zeta potential data is commonly to characterize the stability of colloid solution. In addition, the zeta potential is adopted to assess the surface

charge of nanoparticles (*J. Control. Release. 2016, 235, 337-351*). It is very common in previous literature to use the changed zeta potential to prove that nanoparticles are successfully modified.

For example:

- 1) *Adv. Mater. 2019, 31, 1905271*: “The zeta potential of CMS changed from 2.62 to -13.7 mV after PEGylation”

Figure R1. The zeta potential of CMS, PEG-CMS and PEG-CMS@GOx.

- 2) *Nano Lett. 2020, 20, 4177–4187*: “The evolution of the size and zeta potential in the processing of nanobowls and DOX@NbLipo was monitored at each step”

Figure R2. Evolution of zeta potential in the process of preparing DOX@NbLipo.

In our manuscript, the zeta potential of Fe₃O₄/CNT changed from -13.145 to $+21.015$ mV after loading Gent and PCM, which proved the successful loading of Gent and PCM in Fe₃O₄/CNT.

On this basis, the manuscript has been modified as followed:

In page 8, we made a revision “...The zeta potential of Fe₃O₄/CNT (**Supplementary Fig. 7**) changed from -13.145 to +21.015 mV after loading Gent and PCM...”

Comment 6: Fe₃O₄ or CNT alone has no microwaveocalortic effect, but Fe₃O₄/CNT has microwaveocalortic effect. This should be addressed or discussed in depth, and the heat conversion efficiency of the microwave-excited Fe₃O₄/CNT needs to be provided.

Reply: Thank you very much for the professional suggestion. Fe₃O₄/CNT has higher microwaveocalortic effect than Fe₃O₄ or CNT alone due to the good impedance matching and reasonable attenuation constant (mainly includes heterointerfaces polarization and promoted multiple scattering), and the microwaveocalortic conversion efficiency (η) of Fe₃O₄/CNT dispersed in saline solution is determined to be 35.7%. The corresponding calculation is as following (supplemented in Page S31):

From an energy balance of the system, the microwaveocalortic transduction efficiency can be calculated⁵. The total energy for the system is as follows:

$$Q_{input} \times \eta = Q_{heat} + Q_{loss} \quad (1)$$

Q_{input} is the energy from the MW source. Q_{heat} is the temperature of Fe₃O₄/CNT solution. Q_{loss} is the heat loss to the surroundings. η is defined as nominal microwaveocalortic conversion efficiency.

$$Q_{input} = p \times s_1 \times (t_n - t_0) \quad (2)$$

Where p is output power, s_1 is the heating surface area of the container, t_n is the highest temperature of the steady state, t_0 is the initial temperature before MW heating.

$$C_{0.9\%NaCl} = A_{water} \times C_{water} + A_{15\%NaCl} \times C_{15\%NaCl} \quad (3)$$

where A_{water} and $A_{15\%NaCl}$ are volume-weight of the water and 15% NaCl solution, respectively. C_{water} and $C_{15\%NaCl}$ are the specific heat capacity of the water and 15% NaCl solution, respectively.

$$\begin{aligned} Q_{heat} &= \sum_{t=t_0}^{t=t_n} C_{0.9\%NaCl} \times m \times \Delta T \\ &= C_{0.9\%NaCl} \times m \times (T_n - T_0) \end{aligned} \quad (4)$$

$$\begin{aligned} Q_{loss} &= \sum_{t=t'_0}^{t=t'_n} h \times s_1 \times \Delta (T_{Fe_3O_4/CNT} - T_{Env}) \times \Delta t \\ &= \sum_{t=t'_0}^{t=t'_n} h \times s_1 \times \Delta (f_1(t) - f_2(t)) \times \Delta t \end{aligned} \quad (5)$$

Where h is the heat transfer coefficient, $T_{Fe_3O_4/CNT}$ is the temperature of Fe₃O₄/CNT solution. T_{Env} is the ambient surrounding temperature, $f_1(t)$ and $f_2(t)$ are the function formulas of $T_{Fe_3O_4/CNT}$ and T_{Env} for time. According to the definition of integral, the calculation equation can be transformed as follows:

$$\begin{aligned}
Q_{loss} &= \sum_{t=t'_0}^{t=t'_n} h \times s_1 \times \Delta(f_1(t) - f_2(t)) \times \Delta t \\
&= h \times s_1 \times \sum_{t=t'_0}^{t=t'_n} [(f_1(t) - f_2(t))] d_t \\
&= h \times s_1 \times \int_{t'_0}^{t'_n} [(f_1(t) - f_2(t))] d_t \\
&= h \times s_1 \times \Delta S
\end{aligned} \tag{6}$$

$h \times s_1$ is determined by measuring the process of the free heat dissipation stage II after withdrawing the microwave source.

When the solution reached stable states:

$$\sum_{t=t'_0}^{t=t'_n} C_{0.9\% Nacl} \times m \times \frac{dT}{dt} = -h \times s_1 \times \Delta T \tag{7}$$

Supposing: $\theta = \frac{\Delta T}{T_{max}}$

$$\begin{aligned}
\sum_{t=t'_0}^{t=t'_n} C_{0.9\% Nacl} \times m \times \frac{d\theta}{dt} &= -h \times s_1 \times \Delta T \\
dt &= -\frac{\sum_{t=t'_0}^{t=t'_n} C_{0.9\% Nacl} \times m \times d\theta}{h \times s_1 \times \theta} \\
\xi_s &= \frac{\sum_{t=t'_0}^{t=t'_n} C_{0.9\% Nacl} \times m}{h \times s_1} \\
t &= \xi_s(-\ln\theta) + b
\end{aligned} \tag{8}$$

And $h \times s_1$ can be calculated by the slope of t vs $-\ln\theta$

The time variation value is assumed to infinite approach zero:

$$\lim \Delta t = \lim(t_n - t_{n-1}) \rightarrow 0^+$$

Finally, η can be calculated as follows:

$$\begin{aligned}
\eta &= \frac{\sum_{t=t_0}^{t=t_n} C_{0.9\% Nacl} \times m \times \Delta T + Q_{loss}}{Q_{input}} \\
&= C_{0.9\% Nacl} \times m \times (T_n - T_0) + \frac{\sum_{t=t_0}^{t=t_n} C_{0.9\% Nacl} \times m}{\xi_s} \times \int_{t_0}^{t_n} [(f_1(t) - f_2(t))] d_t / p \times s_1 (t_n - t_0)
\end{aligned} \tag{9}$$

The calculation results are shown in **Supplementary Table 1**:

Supplementary Table 1 Parameters for calculating microwavecalortic conversion efficiency η of Fe_3O_4/CNT .

$C_{0.9\%Nacl} \times m \times \Delta T$	ζ_s	Q_{loss}	Q_{input}	η (%)
230.81	5.73	40.32	420.18	35.7%

where m is the mass of the 0.9% NaCl solution and ΔT is the temperature increase. Q_{input} is the energy from the MW source. Q_{loss} is the heat loss to the surroundings, and η is defined as MW thermal conversion efficiency. $C_{0.9\%NaCl}$ is the specific heat capacity of the 0.9% NaCl solution.

To better express this point, we have modified the manuscript, as followed:

In Page S31, we added the parameters for calculating microwavecaloric conversion efficiency η of Fe₃O₄/CNT as **Supplementary Table 1** and the corresponding calculation process.

In Page 10, we added “... Furthermore, the microwavecaloric conversion efficiency (η) of Fe₃O₄/CNT dispersed in saline solution is 35.7% (**Supplementary Table 1**)....”

In Page 14, we added: “... To summarize, Fe₃O₄/CNT has higher microwavecaloric effect than Fe₃O₄ or CNT alone due to the good impedance matching and reasonable attenuation constant (mainly includes heterointerfaces polarization and promoted multiple scattering). Specifically, the good impedance matching and interfacial polarization promoted multiple scattering, conductive loss, dipole polarization, and natural ferromagnetic resonance loss, thereby greatly enhancing the microwavecaloric performance of the Fe₃O₄/CNT....”

Comment 7: Page 9 Line 162-168, the author points out that Fe₃O₄/CNT/Gent has degradation characteristics, but it can be observed from the SEM of supplementary Fig. 10 that only the morphology of Fe₃O₄/CNT/Gent has changed. Degradation is not Disintegration? It's a change in shape?

Reply: Thank you very much for the professional suggestion. We are very sorry for our negligence of confusing the basic concepts of Degradation and Disintegration. Indeed, this is the disintegration of Fe₃O₄/CNT/Gent. Before and after the “degradation experiment”, the shape (spherical), crystal structure and composition of Fe₃O₄/CNT/Gent (**Supplementary Fig. 10d**) did not show obvious change, only the particle size changed from the original 110 nm to 50 nm and the solution colour became lighter (may be due to increased solubility) (**Supplementary Fig. 10c**). These results indicate that the *in vitro* “degradation experiment” under our experimental conditions is mainly the disintegration of Fe₃O₄/CNT/Gent.

Especially, previous reports have reported that both Fe₃O₄ and CNT alone can be degraded *in vivo*, the Fe₃O₄ nanoparticles are processed by cells as part of the physiological iron metabolism (*Biochim. Biophys. Acta-Gen. Subj.* 2011, 1810, 361–373.) and the CNT degraded by neutrophil myeloperoxidase (*Nat. Nanotechnol.* 2010, 5, 354-359.), so the final degradability of Fe₃O₄/CNT/Gent *in vivo* is beyond doubt. The disintegrability can favor the nanocapturer to be cleared from the body in a reasonable time once it has fulfilled its therapeutic functions *in vivo*.

Supplementary Fig. 10d. The XRD patterns of Fe₃O₄/CNT/Gent in PBS with shaking at 37°C for 0 and 56 days.

Supplementary Fig. 10c. Fe₃O₄/CNT/Gent size distribution in PBS with shaking at 37°C for 0, 7, 28, and 56 days.

To better express this point, we have modified the manuscript, as followed:

In Page 9, we added “...And the biodegradation behavior of Fe₃O₄/CNT/Gent is discussed in the **Supplementary Fig. 10.....**”

In Page S11, we added **Supplementary Fig. 10c and Supplementary Fig. 10d.**

In page S11, we made a revision “...As shown in **Supplementary Fig. 10a, b**, compared to Day 0, the Fe₃O₄/CNT/Gent at Day 7 exhibited almost the same absorption curve. Over time, the absorbance of the Fe₃O₄/CNT/Gent gradually reduced; after immersion in PBS for 56 days, it decreased sharply, with

obvious color fading and the particle size changed from the original 110 nm to 50 nm (**Supplementary Fig. 10c**), but crystal structure and composition of Fe₃O₄/CNT/Gent did not show obvious change (**Supplementary Fig. 10d**), indicating the gradual disintegrability of Fe₃O₄/CNT/Gent *in vitro* under our experimental conditions. Especially, previous reports have reported that both Fe₃O₄ and CNT alone can be degraded *in vivo*, i.e., the Fe₃O₄ nanoparticles are processed by cells as part of the physiological iron metabolism³ and the CNT degraded by neutrophil myeloperoxidase⁴, so the degradability of Fe₃O₄/CNT/Gent *in vivo* is beyond doubt. The disintegrability can favor nanocapturer to be cleared from the body in a reasonable time once it has fulfilled its therapeutic functions *in vivo*.....”

Comment 8: In the manuscript, the author only gives the characterization data of microwave-excitation Fe₃O₄/CNT/Gent release Gent, and should also provide the Gent release curve of Fe₃O₄/CNT/Gent without microwave-excitation.

Reply: Thank you very much for the professional suggestion. Following your advice, we have supplemented the Gent release curve of Fe₃O₄/CNT/Gent without microwave-excitation in **Fig. 3b**, and the corresponding descriptions have been shown in **Page 10**: “...In contrast, the nanocapturer released little Gent (31.6%) without MV treatment (Ctrl) after leached 48 hours, which was observed from the platform of the leaching curve...”

Fig. 3b In vitro Gent release profile from Fe₃O₄/CNT/Gent under MV excitation or not.

Comment 9: Supplementary Fig. 11 showed that Fe₃O₄/CNT/Gent have excellent heating effect in 5 min under microwave-excitation. Is it not too long to choose 20 min in vivo experiment?

Reply: Thank you very much for the professional suggestion. As the reviewer mentioned, Fe₃O₄/CNT/Gent have excellent heating effect in 5 minutes under microwave excitation. However, killing

MRSA needs to be maintained at high temperature for a certain period of time. In addition, the antibacterial rates of Fe₃O₄/CNT/Gent for MRSA after heating for 5, 10, 15 and 20 minutes were 6.96%, 47.82%, 72.68% and 99.72%, respectively (**Supplementary Fig. 18**). The high antibacterial rate makes 20 minutes stand out. Thus, we chose 20 min *in vivo* experiment. Notably, *in vivo* experiment, the temperature of Fe₃O₄/CNT/Gent was higher than 50 °C after MV-irradiated for 5 minutes and then maintained < 55 °C for another 15 minutes by controlling the MV power. It is not that the temperature can rise uncontrollably within 20 minutes for achieving minimal invasion of normal tissues and highly effective bacterial destruction within a short time (*Nat. Commun.* 2019, 10, 4490).

Supplementary Fig. 18 MRSA mixed with Fe₃O₄/CNT/Gent (1 mg mL⁻¹), then exposed to MV for different times (5, 10, 15 and 20 minutes) spread onto LB agar plates and incubated at 37 °C for 20 hours.

To better express this point, we have modified the manuscript, as followed:

In Page S20, we added **Supplementary Fig. 18**.

In Page 18, we added “...In addition, the antibacterial rates of Fe₃O₄/CNT/Gent for MRSA after heating for 5, 10, 15 and 20 minutes were 6.96%, 47.82%, 72.68% and 99.72%, respectively (**Supplementary Fig. 18**). Thus, 20 min MV irradiation was chosen for *in vivo* experiment. Notably, the temperature of Fe₃O₄/CNT/Gent was higher than 50°C after MV-irradiation for 5 minutes and then maintained below 55 °C for another 15 minutes. During this course, the temperature is adjustable by controlling the MV power, so that minimal invasion of normal tissues and highly effective bacteria-killing can be achieved in a short time¹².....”

Comment 10: Word space missing throughout the article. Kindly check.

Reply: Thank you very much for your kind reminder. We are very sorry for our negligence and we have carefully revised our manuscript following your advice.

Comment 11: There are some grammatical errors in this paper, so it is suggested to be polished by a native speaker before being published.

Reply: Thank you very much for the professional suggestion. We are very sorry for our negligence and we have carefully revised our manuscript following your advice. The English language of this paper was edited by a senior Editor from Scribendi Inc., a Canada company that was founded in 1997 by Chandra Clarke and Terry Johnson to connect ideas and people worldwide through professional language services (<https://www.scribendi.com/about.en.html>).

Comment 12: References are incomplete.

Reply: We are very sorry for our negligence and we have carefully revised our manuscript following your advice. We added the corresponding literature references in page 4 of the revised manuscript, as following (supplemented in Page S38):

26. Zhao T, *et al.* Surface-kinetics mediated mesoporous multipods for enhanced bacterial adhesion and inhibition. *Nature Communications* **10**, 4387 (2019).

Response to Reviewer 2#

Original Comment: In the manuscript titled “A microwave-excited nanocapturer with magnetic targeting to precisely eradicate MRSA-infected osteomyelitis”, Qiao et al. designed microwave-responsive system composed of Gent-loaded $\text{Fe}_3\text{O}_4/\text{CNT}$, which can be employed as intelligent antibacterial nanocapturer and microwave-responsive bacteria-killing agent. Their results show that this system can achieve not only precise bacteria-targeting but also magnetic targeting, thus resulting in high MRSA-binding over 95% and restricting the spread of bacteria through blood. This is very important for clinical application. Their further antibacterial experiments disclosed that this synthesized nanocapturer possessed highly effective bacteria-killing efficacy against MRSA and *Escherichia coli* within 20 min microwave irradiation, which was ascribed to the synergistic action of microwaveocaloric and chemo therapy.

Totally, this is a quite interesting paper, which is scientifically organized. The experimental design is very reasonable and feasible. The large amount of experimental data sufficiently support their findings. I believe that this work will bring a new insight for developing novel microwave-responsive materials for rapid and effective treatment of bacteria (including MRSA and *E. coli*) infected diseases in deep tissues. Therefore, I enthusiastically recommend publication of this paper in Nature Communications after addressing the following minor issues:

Reply: We would like to thank the reviewer for his/her positive recommendation and invaluable comments and suggestions.

Comment 1: The author clearly analyses the reasons why $\text{Fe}_3\text{O}_4/\text{CNT}/\text{Gent}$ has different capture capabilities for Gram-negative bacteria and Gram-positive bacteria. Have you considered the effect of electrostatic adsorption on the capture effect? It is recommended that the author analyse this.

Reply: Yes, we have considered this issue. We excluded the electrostatic adsorption as the main reason of $\text{Fe}_3\text{O}_4/\text{CNT}/\text{Gent}$ for capturing bacteria as following reasons: Both the bacteria and the $\text{Fe}_3\text{O}_4/\text{CNT}$ are all negatively charged and the Fe_3O_4 is positively charged. If the binding capacity is due to electrostatic bonding, the Fe_3O_4 should be more easily binding with bacteria rather than the non-binding ability shown in Fig. 4a and c.

To better illustrate this point, we have modified the manuscript, as followed:

In Page 14, we added “...First, we ruled out electrostatics as the main origin for bacteria binding, because both the bacteria and the $\text{Fe}_3\text{O}_4/\text{CNT}$ are all negatively charged. Furthermore, if it is due to electrostatic bonding, the Fe_3O_4 which is positively charged should be more easily binding with bacteria rather than the non-binding ability shown in **Fig. 4a** and **c**....”

Comment 2: The author used different thicknesses of pork (0, 8, 10, 12, and 16 mm) to prove the strong microwave penetration and recorded the corresponding temperature, but there is no calculation about the mortality rate of bacteria, it is recommended to add bacterial live death staining under different thicknesses of pork, in order to evaluate the deep sterilization effect of microwave.

Reply: For evaluate the deep sterilization effect of $\text{Fe}_3\text{O}_4/\text{CNT}/\text{Gent}$ under microwave, a live-dead fluorescence staining experiment was performed. And, under 0.1 W cm^{-2} MV excitation the efficiency of anti-MRSA at thicknesses of 0, 8 and 10 mm was calculated to be 100%, 98.55% and 86.99%, respectively. At 12 and 16 mm thickness, the antibacterial rate is less than 10% imputing to the temperature decrease caused by the thickness of the penetrated tissue increases. Fortunately, we can control the temperature through the power of the microwave (**Fig. 5d**), which means we can control the penetration thickness by adjusting the microwave power.

Supplementary Fig.19 Live-dead fluorescence staining of MRSA after MV (0.1 W cm^{-2}) excitation at different pork thickness (0, 8, 10, 12, and 16 mm). Green fluorescence was stained by SYTO9 dye, which indicated live bacteria; Red fluorescence was stained by PI dye, which indicated dead bacteria. Scale bar, $10 \mu\text{m}$.

To better express this point, we have modified the manuscript, as followed:

In Page S21, we added the Live-dead fluorescence staining as **Supplementary Fig.19**.

In page 20, made a revision: “...we investigated the tissue penetration depth of the Fe₃O₄/CNT/Gent for microwavecaloric conversion and anti-MRSA behaviors under MV irradiation...”

In page 20, we added: “... Especially, to observe the antibacterial activities of Fe₃O₄/CNT/Gent solution with different pork thickness, fluorescence staining was carried out (**Supplementary Fig.19**), and the antibacterial efficacy of Fe₃O₄/CNT/Gent was calculated to be 100%, 98.55% and 86.99%, respectively under the pork thickness of 0, 8 and 10 mm after the microwave (0.1 W cm⁻², 20 min) irradiation. In contrast, under the thickness of 12 and 16 mm, the antibacterial rate was less than 10% under the same MV irradiation, which was ascribed to the temperature drop as the thickness of the penetrating tissue increased.....”

Comment 3: It is very important to systematically evaluate the basic cytotoxicity of the used antibacterial agent Fe₃O₄/CNT/Gent. In this manuscript, only MC3T3-E1 osteoblast was used for evaluation. It is recommended to supplement its toxicity to connective tissues such as fibroblasts.

Reply: The NIH3T3 cell line is a widely used fibroblast cell line. We selected NIH-3T3 (ATCC CRL-1658) fibroblasts to supplement the Fe₃O₄/CNT/Gent cytotoxicity test.

After one and five days of coculturing, the Fe₃O₄/CNT/Gent also exhibited negligible cytotoxicity in the concentration range of 0.1–1 mg mL⁻¹.

Supplementary Fig. 21 d,e, The viability of NIH-3T3 cells cocultured with different concentrations of Fe₃O₄/CNT/Gent after coculturing for one (**d**) and five (**e**) days.

On this basis, the manuscript has been modified as followed:

In page S23, we added the viability of NIH-3T3 cells cocultured with different concentrations of Fe₃O₄/CNT/Gent as **Supplementary Fig. 21 d,e** with the description “... Similarly, for NIH-3T3

fibroblasts, with the increase of Fe₃O₄/CNT/Gent concentration, the cell viability did not decrease significantly after one and five days of co-cultivation, all of which were above 80%....”

Comment 4: The microwaveocaloric process includes the interaction between activated carrier/dipoles and microwave field. Some works devoted to analyzing polarization field and carrier motion behavior, such as Adv. Energy Mater. 2019, 9, 1902174 and Adv. Energy Mater. 2020, 1, 1904072, can help the author to thoroughly describe the related mechanism.

Reply: We would like to thank the referee for this valuable comment, which offers us a better understanding of the microwaveocaloric analysis of Fe₃O₄/CNT. Thus, in our revised paper, these valuable literatures have been cited in page 3 of the revised manuscript, as following (the numbers are their locations in the reference list in the manuscript):

“... 16. Jiao W, *et al.* Yolk-Shell Fe/Fe₄N@Pd/C Magnetic Nanocomposite as an Efficient Recyclable ORR Electrocatalyst and SERS Substrate. *Small* **15**, e1805032 (2019).

17. Jiao W, *et al.* Hollow Palladium□Gold Nanochains with Periodic Concave Structures as Superior ORR Electrocatalysts and Highly Efficient SERS Substrates. *Advanced Energy Materials* **10**, 1904072 (2020).

18. Yang L, *et al.* Conductive Copper Niobate: Superior Li⁺□Storage Capability and Novel Li⁺□Transport Mechanism. *Advanced Energy Materials* **9**, 1902174 (2019).....” in the manuscript.

In page 3, we supplemented a statement “... Che et al. developed the enhanced coupling of the electronic field to facilitate the chemical activity of the magnetic nanoparticles^{16,17} and pioneer the investigation of chemical reaction mechanisms in the nanometer devices¹⁸.....”

Lastly, we would like to thank the Editor and all the Reviewers again for their time and effort in helping us improve the quality of this manuscript. It is greatly appreciated. We hope that our responses are satisfactory and the revised manuscript could meet the standard of *Nature Communications*.

Reviewers' Comments:

Reviewer #1:

Remarks to the Author:

I consider that the issues presented by Referees have been well addressed to greatly improve the original manuscript. Thus, I believe that the manuscript is now on a suitable level to be published in NC.

Reviewer #2:

Remarks to the Author:

Recommendation: Accept.

Comment: The authors have well addressed all the issues and improved the quality of this manuscript according to the proposed suggestions. Therefore, I recommend the publication of this paper in Nature Communications.